# Spatiotemporal Data-Driven Multiperiod Relocation Optimization of Emergency Medical Services: Maximum Equality Objective

Xinxin Zhou [1,2,3], Yujie Chen [1], Yingying Li [2,3,4], Bingjie Liu [2,3,4] and Zhaoyuan Yu [2,3,4],*

1 School of Geographic and Biologic Information, Nanjing University of Posts and Telecommunications, Nanjing 210023, China; zhouxinxin@njupt.edu.cn (X.Z.); chenyujie@njupt.edu.cn (Y.C.)
2 Key Laboratory of Virtual Geographic Environment, Ministry of Education, Nanjing Normal University, Nanjing 210023, China; 221302181@njnu.edu.cn (Y.L.); 201345011@njnu.edu.cn (B.L.)
3 Jiangsu Center for Collaborative Innovation in Geographical Information Resource Development and Application, Nanjing 210023, China
4 College of Geographical Science, Nanjing Normal University, Nanjing 210023, China
* Correspondence: yuzhaoyuan@njnu.edu.cn

**Abstract:** As a kind of first aid healthcare service, emergency medical services (EMSs) present high spatiotemporal sensitivity due to significant changes in the time-dependent urban environment. Taking full advantage of big spatiotemporal data to realize multiperiod relocation optimization of EMSs can reduce idle resources and improve service utilization efficiency and fairness. First, we established the dynamic time-dependent accessibility and equality model to formulate the multiperiod maximization objective of global equality. Second, we proposed a capacitated integer evolution algorithm that relocates emergency medical vehicles to optimize the scheduling scheme. Based on multiperiod mobile phone records and multiperiod online route planner data, the equality of EMSs in the research metropolis, Nanjing, China, rose by 41.5% on average, which has an incentivizing effect on alleviating the tension of prehospital service and minimizes accessibility disparities without constructing more infrastructure. We also created maps to visualize the changes in equality patterns over time. This relocation optimization approach can be regarded as a trade-off approach to dispatch time-dependent sensitive services and provide a practical tool for healthcare decision-makers to evaluate public healthcare systems and improve strategic urban service planning.

**Keywords:** emergency medical services; accessibility; equality; relocation optimization; health services; spatiotemporal analysis

## 1. Background

In recent decades, economists and geographers have paid considerable attention to geo-health topics associated with the human geographical living environment around the world [1]. As a key component of healthcare services, emergency medical services (EMSs)—including EMS stations and emergency medical vehicles (EMVs)—provide much-needed prehospital emergency medical care to patients before they are transferred to a hospital [2,3]. EMSs have a high spatiotemporal sensitivity among various health facilities, one example being the golden eight minutes. In most cases, EMSs face a dilemma characterized by insufficient resources and a large gap between response time and prime time, which largely increases the difficulty and workload when the emergency department implements policies [4].

One primary problem with the insufficient and unbalanced use of EMS resources is that the current scheduling strategy fails to implement dynamic load balancing at each EMS station in accordance with the actual dynamic demand [5,6]. The main reasons are as follows.

(1) **Simulated urban data.** In EMS location and relocation studies, simplifying assumptions are used in the modeling phase to assess the models analytically [7]. Limited by the availability of data and the effectiveness of data processing methods, city managers mainly employ population data based on statistical surveys or Euclidean distance traffic data based on GIS software simulations when compiling EMS planning and dispatch strategies. To some extent, the generalized data representation of urban space loses the actual activity features of urban residents. It is crucial to conduct a quantitative evaluation of the realistic urban environment based on big geospatial data.

(2) **Static urban environment.** Traditional research on the spatial accessibility of EMSs lacks consideration of spatiotemporal dynamics and spatiotemporal heterogeneity, which may lead to a problem regarding uncertain geographic context [8]. In the early stages, EMS location studies assumed that the urban population distribution was static and the urban traffic commuting state was consistent. Scholars first researched a series of stationary models for EMV locations, met the requirements for maximum coverage under limited EMV resources, and decreased the allocation and operating costs [9]. Dynamically estimating potential emergency demands based on big spatiotemporal data is a critical prerequisite for rationally dispatching EMVs at each station and shortening the response time.

(3) **Equilibrium between equality and efficiency is challenging.** The objective of EMSs is to obtain emergency resources in the shortest time [6]. If the response time of EMSs can be shortened, the survival rate of patients will be improved. Inequitable solutions can be generated by classic shortest-time models due to imbalances in economic level between cities and densely populated and low-populated areas [10]. Efficiency disparities exist, especially in medical facilities in developing countries [11]. One important concept for people is equity and fairness, the "equalization of basic public services", which is mutually affected by and is related to the degree of satisfaction regarding how people's needs are met based on available resources [12].

The rapid development of big spatiotemporal data has led to great opportunities to solve sustainable problems affecting cities and the environment [13]. With the development of wireless positioning and communication technology, large-scale dynamic traffic information can be collected for EMSs, making it possible to study the accessibility of actual road traffic conditions. Mobile phone records [14], social media data (such as Twitter data) [4], web map service data (such as online route planner data provided by Google Maps API) [15], and trajectory data of ambulance trips [7] are widely used in the field of EMS optimization. Therefore, the study of optimizing EMS relocation based on big spatiotemporal data has great practical utility without raising investment in the scale of EMS infrastructure.

This research attempts to reconcile the discrepancy between demand and availability for EMSs in various time slots. We provide a framework for the multiperiod relocation optimization of emergency medical services (MRO-EMS) based on big spatiotemporal data under the limitations of the constrained capacity of EMVs. We preprocess the time-dependent urban data based on multiperiod mobile phone records (M-MPR) and multiperiod online route planner (M-ORP) data. The evaluation and optimization EMS model is established to identify the spatiotemporal accessibility and equality of EMSs, and a relocating optimization algorithm is implemented to schedule different emergency moments, thereby improving prehospital emergency medical care and maximizing service resources. The potential contributions of our research are as follows:

(1) **Realistic urban spatiotemporal data.** We adopted a novel approach for utilizing route planner data from web maps and city-wide fine-grained urban population distributions over multiple periods to address the issues of EMS accessibility evaluation and relocation. The EMS model was implemented in the megacity of Nanjing as a case study, and spatiotemporal patterns were discussed.

(2) **Dynamic time-dependent urban environment.** EMS demand varies over space and time. Demand patterns change, and contingencies occur occasionally, which is why healthcare emergency operations can be delayed [16]. The MRO-EMS method can combine the demand status of the urban population with the traffic environment over various periods, redeploy and relocate idle EMVs between stations, transfer EMVs from potentially low-demand stations to high-demand stations, and realize scheduling changes such that EMVs are not fixed to a specific station in a dynamic environment, ensuring that the call demand near high-demand stations can be promptly responded to.

(3) **Toward maximum equality objective.** An improved equality model considering dynamic time dependence is proposed based on the model to analyze spatiotemporal data. The dynamic relocation algorithm of EMVs between EMS stations is a viable, practical, and relatively straightforward operation that is anticipated to provide considerable effects given the fundamental fact of the lack of service.

The remainder of this paper is structured as follows. Sections 2 and 3 introduce related work and the methodology, respectively. Section 4 briefly introduces the study area and data. The experiments and results are shown in Section 5. Finally, we end this study with a brief discussion and conclusion in Section 6.

## 2. Related Work

EMS location models can be categorized into three classes: static class, probabilistic class, and dynamic class [17]. The first class focuses on earlier emergency location coverage models; the probabilistic class defines the EMV unavailability ratio; the last class focuses on depicting the means of reassigning vehicles dynamically. In the early stages, scholars researched a series of stationary models for EMV locations, met the requirements for maximum coverage under limited EMS resources, and decreased the allocation and operating costs. Such models include the location set covering model, maximal covering location problem, tandem equipment allocation model, backup coverage problem, and double standard model. Based on the static model, scholars introduced the concept of probability coverage and coverage at different times [18]. However, all the cases described view EMS location as a static problem, meaning EMVs have immovable location sites and fixed numbers across the timeline. The perceived quality of EMSs would be degraded as a result [1,19].

To remedy the shortcomings of static models, researchers devised dynamic EMS localization. For example, the fleet's location can be adjusted in real time if some EMVs become occupied [20,21]. Considerable attention has been devoted to the development of approaches to solve both multiperiod and dynamic relocation problems. New studies continue to appear regularly, such as TIMEXCLP [22], DDSMt [23], DACL [24], mDSM [25], MPBDCM [26], and time-dependent MEXCLP [27] (TIMEXCLP refers to the maximal expected coverage location model with time variation; DDSMt refers to the dynamic double standard model; DACL refers to the dynamic available coverage location model; MPBDCM stands for the multiperiod backup double covering model; mDSM represents the multiperiod double standard model). However, the existing research relies mainly on simulated, simplified data of the urban environment to realize the optimal solution rather than big geodata following the urban real-time-dependent environment. Additionally, the relocation of EMVs is a requirement for real-time management of the dynamic allocation model, which is bound to bring about high costs and heavy workloads for EMV staff [4].

Geospatial big data represent opportunities for scientific research and provide new transformation paradigms for different disciplines, especially at the intersection of a wide range of disciplines, such as humanities, physical sciences, and engineering [28]. Abundant information, such as real-time traffic conditions and real-time population distribution, is increasingly accessible due to diverse information and communication technology (ICT) connections [29]. Fortunately, faster heuristics and enhanced computer technologies enable EMV location issues to be solved in real time, and enable a new EMV deployment strategy

to be computed at any time using real-time data [19]. Van Barneveld, et al. [30] developed a novel perception of optimizing the redistribution of EMVs using trace-driven simulations based on a real dataset from EMV providers in the Netherlands. Enayati, et al. [31] took advantage of a discrete-event simulator developed for a large real dataset to estimate the capability of the timely approach compared with that of two benchmarks. Nilsang, Yuangyai, Cheng and Janjarassuk [4] was devoted to the study of a covering model based on social media analysis that was designed to establish a system for EMV relocation by taking into account real-time data from a social media application (Twitter) involved in communication and managing emergencies. Carvalho, et al. [32] utilized real EMS data from Lisbon, Portugal, to conduct experiments to expand the system's coverage as much as feasibly possible by taking advantage of a time-preparedness measure that allows relocation to any base. Kochetov and Shamray [33] developed a model that simulates the whole work process of the EMS of Vladivostok, in which both the stochasticity of the problem and the changes in road network loads cannot be ignored. Jánošíková, Jankovič, Kvet and Zajacová [7] proposed a mathematical relocation programming model to reduce the average response time of EMVs, which was assessed using real historical data from the Slovak Republic. Hajiali, Teimoury, Rabiee and Delen [16] established a model-driven decision support system that divided and prioritized demand into four categories and updated them continuously based on the current location of idle EMVs.

## 3. Methods

### 3.1. Outline of the Proposed Approach

The main challenge in this research is determining how many EMVs should be relocated to each EMS station at different periods. The proposed MRO-EMS framework is based on big spatiotemporal data (Figure 1). We obtained fine-grained urban population and dynamic traffic data at multiple periods (Section 4.2) based on big spatiotemporal data in the initial stage. Simultaneously, we defined accessibility and equality models (Sections 3.2 and 3.3) aiming at maximum dynamic time-dependent equality. In the optimization stage, we constructed the capacitated integer evolution algorithm (CIEA) (Section 3.4), which has three leading operators: decision variables, relocation plan, and optimization objective. In the output stage, we established EMV relocation schemes for various scenarios and performed multisequential analyses of the results.

### 3.2. Dynamic Time-Dependent Accessibility Model

Accessibility is the research foundation of service equality and spatial optimization [34], and can be measured in two main ways: location-based and individual-based [35]. Traditional location-based accessibility methods include the two-step floating catchment area method (2SFCA), the gravity model, the three-step floating catchment area method (3SFCA), the Huff model, and other models [36]. In recent years, accessibility research has developed from static to multiperiod spatial evaluation, showing time-series dependence. Typical research methods include dynamic accessibility models based on M-MPR and M-ORP data [14], while spatial accessibility focuses on traffic congestion [37]. Wang [36] integrated various accessibility evaluation methods into a unified form so that the advantages of the 2SFCA and the gravity model can be combined in a unified mathematical paradigm. However, the general model with the distance decay function overestimates the potential population demand when multiple providers can be used in a demand location without considering the competitive potential. Combining the potential of competition among providers, Wan, et al. [38] improved the 2SFCA and proposed the 3SFCA. It included an initial step that increases the competitive weight ($G_{ij}$) from the demand point to the provider point within the search radius $d_0$ before the 2SFCA process. The value of $G_{ij}$ is determined by the following equation:

$$G_{ij} = \frac{W_{ij}}{\sum_{k \in \{Dist(i,k) < d_0\}} W_{kj}} \tag{1}$$

where $W_{ij}$ and $W_{kj}$ are the assigned Gaussian weights for location $i$ to service site $j$ and $k$, respectively. $Dist(i, k)$ is the travel cost from $i$ to any service site $k$ within the catchment, and $d_0$ is the catchment size. The detail computation of the Gaussian weights will be discussed in paper [38].

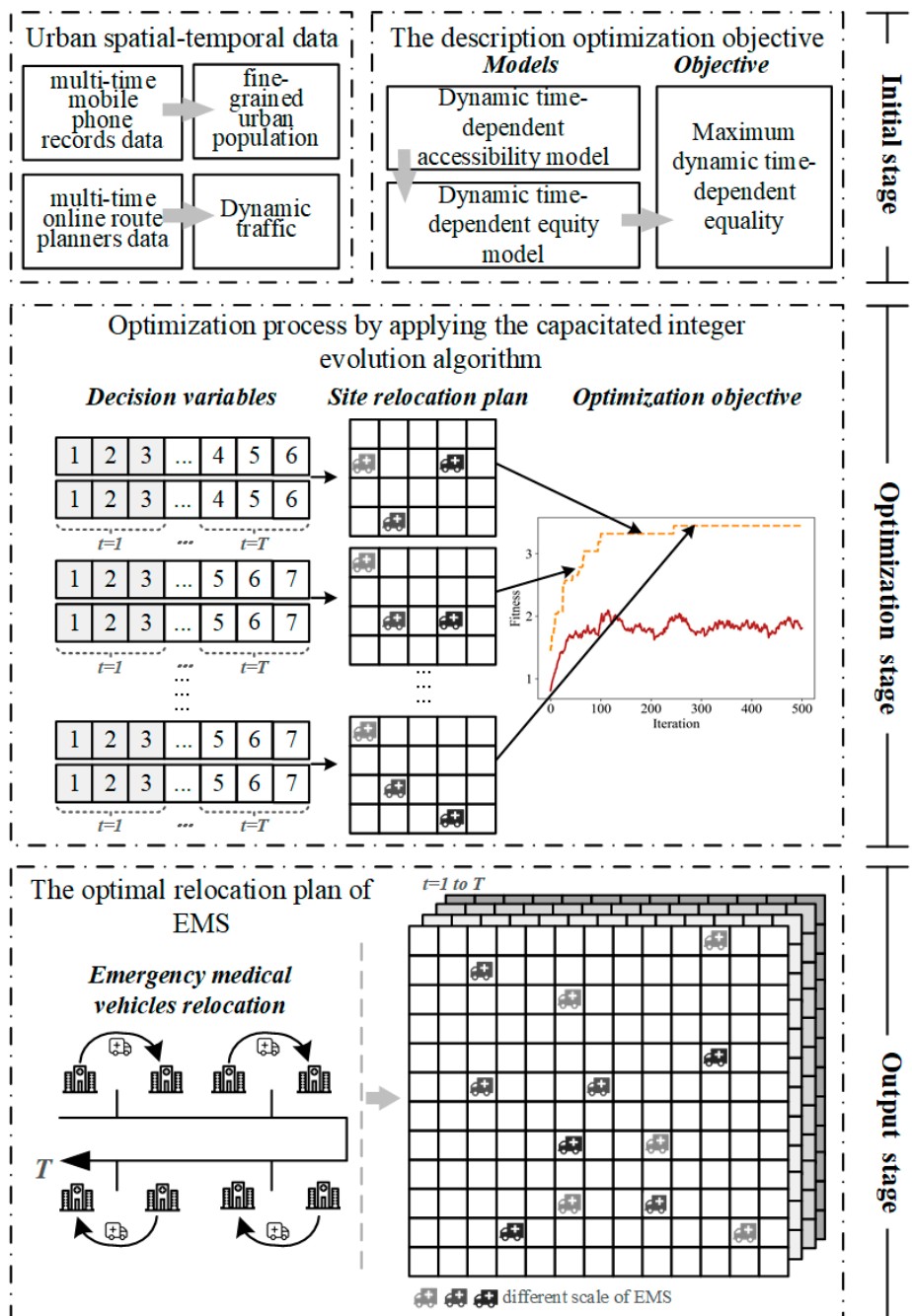

**Figure 1.** Outline of the proposed approach.

Based on 2SFCA and Equation (1), 3SFCA can be written as (Equations (2) and (3)):

$$D_j = \frac{E_j}{\sum_{k=1}^{M} Q_k G_{ij} f\left(d_{kj}\right)} \tag{2}$$

$$A_i = \sum_{j=1}^{N} D_j G_{ij} f\left(d_{ij}\right) \tag{3}$$

where the distance decay function $f(d_{ij})$ in Equations (2) and (3) can be written as:

$$f(d_{ij}) = \begin{cases} d_{ij}^{-\beta}, & d_{ij} \leq d_0 \\ 0, & d_{ij} > d_0 \end{cases}$$ (4)

where

$i$—the index of demand point;

$j$—the index of supply point;

$A_i$—the spatial accessibility of demand point $i$;

$E_j$—the number of pediatricians indicating the service resource supply capacity of provider point $j$;

$D_j$—the supply–demand ratio of service facility $j$;

$Q_k$—the demand amount of demand point $k$, such as population number (unit: 10,000 people, consistent with the dimensions in the previous data description);

$f(d_{ij})$—generalized distance decay function;

$d_{ij}\left(d_{kj}\right)$—the traffic impedance between demand point $i$ and supply point $j$;

$M$—the total number of demand points;

$N$—the total number of supply points;

$d_0$—the travel time threshold; that is, the search radius;

$\beta$—the coefficient of travel friction, set to 1, with reference to Yao, et al. [39].

The 3SFCA model (Equations (2) and (3)) has not yet considered the variation of time slots. After introducing the time slot $t$ into the 3SFCA model, we improved and proposed a dynamic time-dependent accessibility model for different time slots, as shown in Equations (5) and (6):

$$D_j^t = \frac{E_j^t}{\sum_{k=1}^{M} Q_k^t G_{ij}^t f\left(d_{kj}^t\right)}$$ (5)

$$A_i^t = \sum_{j=1}^{N} D_j^t G_{ij}^t f\left(d_{ij}^t\right)$$ (6)

where

$t$—represents different time slot, $t \in (0, T]$;

$A_i^t$—the spatial accessibility of demand $i$ at time slot $t$;

$E_j^t$—the number of pediatricians indicating the service resource supply capacity of provider $j$ at time slot $t$;

$D_j^t$—the supply–demand ratio of service facility $j$ at time slot $t$;

$Q_k^t\left(Q_i^t\right)$—the demand amount of demand $k$ at time slot $t$;

$d_{ij}^t\left(d_{kj}^t\right)$—the traffic impedance between demand point $i$ and supply point $j$ at time slot $t$.

### 3.3. Dynamic Time-Dependent Equality Model

The concepts and metrics of spatial equality in the context of health care are covered in-depth in a recent review [10,40]. Health equality is described in a Robert Wood Johnson Foundation study as where "everyone has a fair and reasonable chance to be as healthy as possible". In our research, the definition emphasizes equitable accessibility rather than outcomes or service consumption, which can be calculated in terms of the maximum deviation, standard deviation, and coefficient of variance [41]. Wang and Tang [42] proposed an optimization model to minimize the difference in accessibility from demand points to provider locations, thereby realizing the maximization of equality—in other words, minimizing inequality. The objective fitness can be expressed as:

$$min \sum_{i=1}^{M} (IE_i)^2$$ (7)

$$IE_i = A_i - a \tag{8}$$

$$a = \sum_{i=1}^{M} \frac{Q_i}{Q} A_i = \frac{E}{Q} \tag{9}$$

where

*a*—the global average value of accessibility;

$IE_i$—the inequality value of demand *i* that has a positive and negative difference. The higher the positive value is, the higher the level of accessibility beyond the global average and the richer the accessibility.

*E*—the total service (provider) supply capacity;

*Q*—the total demand amount.

However, the existing model cannot meet the requirements of multiperiod evaluation, so we improved the model and obtained a new maximum dynamic time-dependent equality fitness within *T* (Equation (10)), the main optimization problem of our article. Additionally, as a penalty variable, the cardinal sign function $sgn(\Delta_t)$ is introduced to make sure that after optimization is better than before optimization, and to prevent fractional failure as shown in Equations (12) and (13):

$$\text{objective}: \qquad max \frac{\partial * \sum_{t=1}^{T} sgn(\Delta_t)}{\sum_{t=1}^{T} \sum_{i=1}^{M} \left(A_i^t - a^t\right)^2} \tag{10}$$

$$a^t = \sum_{i=1}^{M} \frac{Q_i^t}{Q^t} A_i^t = \frac{E^t}{Q^t} \tag{11}$$

$$\text{where}: \qquad \Delta_t = \sum_{i=1}^{M} \left(A_i^t - a^t\right)^2 - \sum_{i=1}^{M} \left(A_i^{t'} - a^{t'}\right)^2, \ t \in [1, T] \tag{12}$$

$$sgn(\Delta_t) = \begin{cases} 1, \Delta_t \geq 0 \\ 0, \Delta_t < 0 \end{cases} \tag{13}$$

$$\text{subject to}: \qquad E^t \text{ is an invariant constant at time slot } t \tag{14}$$

$$E^t = \sum_{j=1}^{N} E_j^t, \forall \, t \in [1, T] \tag{15}$$

$$E_j^t \in [bottom, top], \forall \, j \in [1, N] \tag{16}$$

$$E_j^t \text{ is an integer}, \tag{17}$$

where

$\partial$—a constant that is set to 500 in this research (Through visualization experiments, this value can scale the dimensionless equality value to the visualized value interval);

*bottom*—the lower (bottom) limit at provider *j*;

*top*—the upper (top) limit at provider *j*;

$Q^t$—the total demand amount at time slot *t*;

$E^t$—the total service (provider) resource capacity at time slot *t*;

$t'$—the original initial moment, representing the status quo before optimization;

$A_i^{t'}$—the accessibility value of demand *i* at initial moment $t'$;

$a^t$—the global average value of accessibility at time slot *t*;

$a^{t'}$—the global average value of accessibility at initial moment $t'$;

$\Delta_t$—the variation of the inequality value after the after optimization at time slot *t*.

*3.4. Capacitated Integer Evolution Algorithm (CIEA)*

The redeployment action of the EMS dispatching center assigns EMVs across stations in each time period based on the scheduling scheme determined by the MRO-EMS method.

The MRO-EMS method is a kind of combinatorial optimization problem that is NP-hard. A variety of intelligent evolutionary algorithms are widely used in healthcare facility location and relocation, including heuristic methods to solve the maximum dynamic coverage location model [43], hypercube model, genetic algorithm applied to emergency station location and EMV allocation plan [44,45], stochastic optimization model applied to EMV redeployment [46], Markov model of EMS systems [3], genetic algorithm integrated with EMS simulation model to select station location [47], and mixed-integer linear programming models [19]. Considering that the standard genetic algorithm is an applicable and effective continuous optimization algorithm, we improved the real number coding rule, solution generation, genetic operators, and capacity constraints to satisfy the problem regarding relocated emergency medical vehicles—a typical discrete integer combinatorial optimization problem—and obtained the CIEA methods (Figure 2).

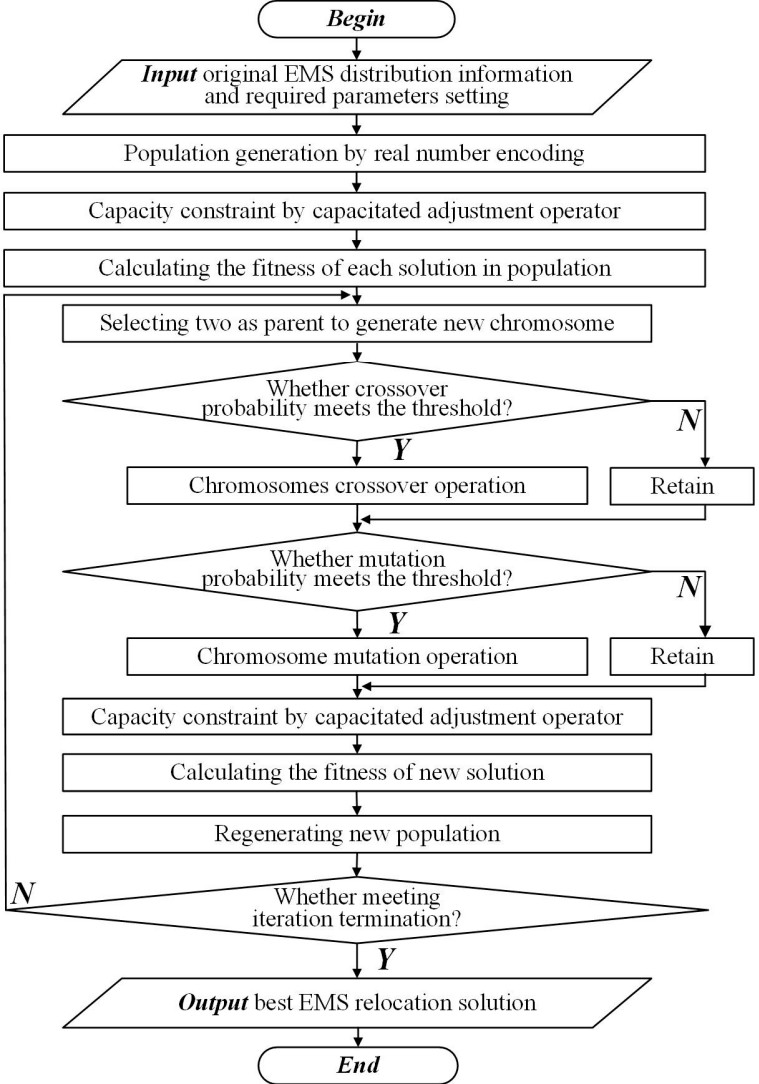

**Figure 2.** Algorithm flow chart of CIEA.

First, the CIEA generates the initial population through a real number encoding mechanism and capacitated adjustment operator. Each solution in the population contains the relocated number of EMVs in every EMS station at four time slots. Second, two chromosomes are selected by the roulette method to perform crossover, mutation, and capacitated adjustment. Third, the newly generated solution replaces the solution of the parent population, and the above operations are repeated. Finally, after a certain number of iterations, the loop is terminated, and the optimal solution is output as the adopted solution. The

optimization objective is to maximize the value of Equation (10), subject to the constraints specified in Equations (14)–(17).

### 3.4.1. Real Number Encoding Mechanism

Real number encoding—the primary step of problem space mapping to the solution space—overcomes the accuracy loss of the traditional binary coding mechanism. Each chromosome solution contains $M * T$ gene positions to save real number encoding results. Each gene position saves the number of EMVs in station $j$ at time $t$. Figure 3a shows a solution example that includes sub-solutions of different time slots based on the real number encoding mechanism.

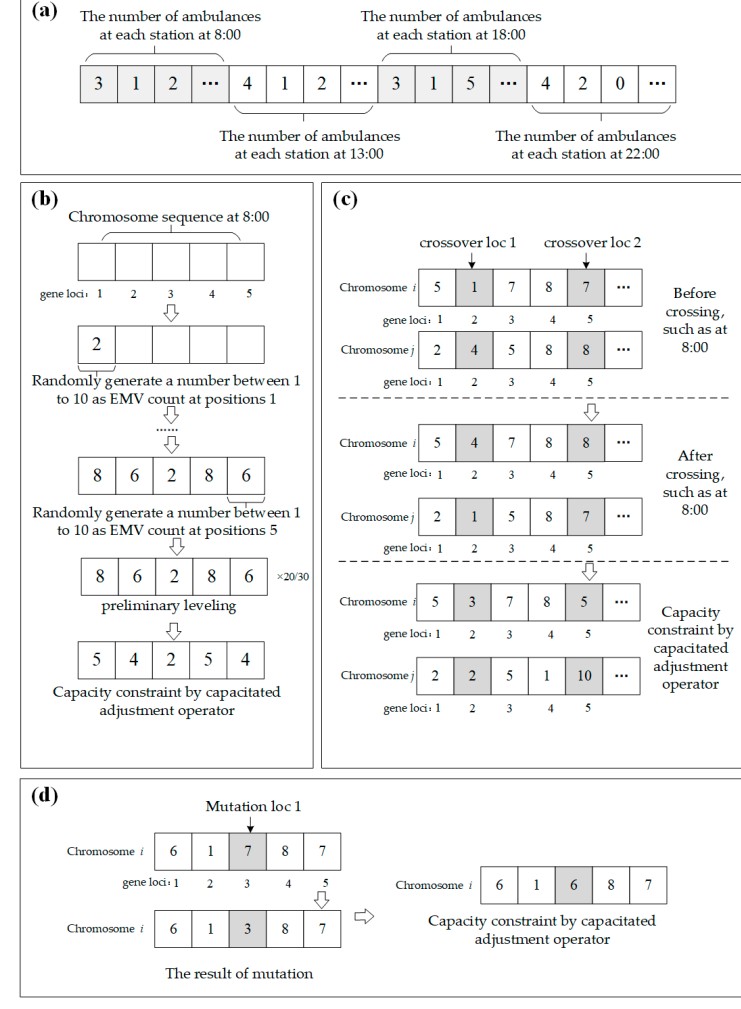

**Figure 3.** The operators' diagrams of CIEA. (**a**) a multiperiod solution example based on real number encoding; (**b**) the chromosome generation example at 8:00, where *bottom* = 1, *top* = 10, *y* = 20, and the number of EMS stations is 5; (**c**) the crossover operation; and (**d**) the mutation operation. To facilitate the description, we consider the crossover operation and mutation operation process of chromosomes at 8 o'clock.

### 3.4.2. Population Generation

Multiple solutions are combined into a solution space population. The steps to generate one solution are as follows (Figure 3b).

**Step 1:** Select station $j$ represented by each gene, and obtain the upper (top) and lower (bottom) limit on the number of EMVs that can be relocated at time $t$;

**Step 2:** Determine whether station $j$ is the last station at the current time slot. If so, terminate the CIEA operation and locate the generation of the chromosome at the next time slot;

otherwise, the number of EMVs refers to a random number $y_i$ between *bottom* and *top* until the chromosome at time $t$ is generated;

**Step 3:** The chromosome generated by the above two steps may exceed the total capacity. Therefore, the value of the chromosome at time $t$ must be preliminarily adjusted. Assuming that the number of EMVs available at time $t$ is $y$ and the number of EMVs is $y_{sum}$, the number of EMVs after adjustment at each position is $y_{i'} = y_i \times y / y_{sum}$;

**Step 4:** Check whether the value of each gene locus is within the *bottom* and *top* limits;

**Step 5:** Use the capacitated integer adjustment operator for further leveling such that the number of individual EMVs satisfies the capacity constraint;

**Step 6:** Continue to generate the values of gene loci at the next time slot and go to **Step 1**.

### 3.4.3. Crossover Operation

Crossover is an essential part of the CIEA in this article. Assuming that the population size is w and the number of crossings between two individuals is $w/2$, the crossing rules are as follows (Figure 3c).

**Step 1:** Two chromosomes ($i$ and $j$) are randomly selected from the population following the roulette method.

**Step 2:** At each time slot, the number of positions where the chromosome crossover operation occurs is $n$. Therefore, the number of crossover gene loci for each chromosome is $n \times T$. A cross gene loci sequence is randomly generated, denoted as $\{X_1, X_2 \ldots X_{nT}\}$, that points to the $\{X_1, X_2 \ldots, X_{nT}\}$ locations of $i$ and $j$, respectively. Assuming that the numbers of EMVs at the intersection of the two chromosomes are $\{i_1, i_2 \ldots i_{nT}\}$ and $\{j_1, j_2 \ldots, j_{nT}\}$, the numbers of EMVs after crossing are $\{i_{1'}, i_{2'} \ldots, i_{nT'}\}$ and $\{j_{1'}, j_{2'} \ldots, j_{nT'}\}$.

**Step 3:** Chromosome $i$ crosses at gene loci $nT$, and the number of EMVs at that gene locus is $i_{nT'} = (i_1 + i_2 + \ldots + i_{nT}) \times j_{nT} / (j_1 + j_2 + \ldots + j_{nT})$, $j_{nT'} = (j_1 + j_2 + \ldots + j_{nT}) \times i_{nT} / (i_1 + i_2 + \ldots + i_{nT})$;

**Step 4:** Repeat the crossing operation until the number of crossings reaches $w/2$.

### 3.4.4. Mutation Operation

To ensure the diversity of chromosomes in the population, some of the chromosomes must undergo mutation operations. The mutation rules are as follows (Figure 3d).

**Step 1:** One chromosome of $i$ is randomly selected from the population following the roulette method;

**Step 2:** The number of chromosome mutation locations at each time slot is $n$. Therefore, the number of individual mutation locations is $n \times T$. A sequence of variant positions is randomly generated. Assuming that the number of EMVs at individual mutation locations is $\{i_1, i_2 \ldots, i_{nT}\}$, and that of the randomly generated mutation sequences is $\{m_1, m_2 \ldots, m_{nT}\}$, the numbers of EMVs after mutation are $\{i_{1'}, i_{2'} \ldots, i_{nT'}\}$, $i_{nT'} = (i_1 + i_2 + \ldots + i_{nT}) \times m_{nT} / (m_1 + m_2 + \ldots + m_{nT})$;

**Step 3:** Repeat steps 1 and 2 until the sum of the gene values at the mutation position is equal to $i_1 + i_2 + \ldots + i_{nT}$.

### 3.4.5. Capacitated Adjustment Operator

The new solution generated by random generation, crossover, and mutation generally does not satisfy the limitation of the capacity scale; thus, it must be repaired. The challenge of the repair process is to ensure that the total number of gene loci is equal to the capacity limitation while also maintaining the number of single gene loci between the upper and lower intervals. The capacitated adjustment operator is used to solve this challenge. The effectiveness of the total capacity and the boundary conditions is commonly disregarded in theoretical research, but determines the method's usefulness in application scenarios. The input parameters of the capacitated adjustment operator contain the upper and lower limits of chromosome length and capacity and the chromosome sequence, and the output result is the capacitated chromosome. The main body of the algorithm is divided into three components: individual initialization, preliminary leveling, and slight leveling. The

specific implementation and the pseudocode of the capacitated adjustment operator are shown in Algorithm 1.

---

**Algorithm 1.** The structure of the capacitated adjustment operator.

1 **Input:** upper limit $UP$, lower limit $BOTTOM$, chromosome dimension $M$, capacity $E\_setting$, chromosome $q_r^t = [x_{r1}^t, x_{r2}^t, \ldots\ldots, x_{rj}^t, \ldots\ldots, x_{rM}^t]$, capacity deviation $\Delta$

2 **Output:** capacitated chromosome $q_r^t$

3 Summing for $q_r^t$ of $[x_{r1}^t, x_{r2}^t, \ldots\ldots, x_{rj}^t, \ldots\ldots, x_{rM}^t]$ and getting SUM value

4 Preliminary leveling:

5 Update each value of $x_{rj}^t$ based on broadcast mechanism $x_{rj}^t = x_{rj}^t * \frac{E\_setting}{SUM}$

6 Slight leveling:

7 **if** $sum([x_{r1}^t, x_{r2}^t, \ldots\ldots, x_{rj}^t, \ldots\ldots, x_{rM}^t]) ! = E\_setting$ **then**

8 $\quad$ $\Delta = sum([x_{r1}^t, x_{r2}^t, \ldots\ldots, x_{rj}^t, \ldots\ldots, x_{rM}^t]) - E\_setting$

9 **end**

10 **while** $\Delta ! = 0$ **do**

11 $\quad$ Randomly selecting the gene postion $u$ of individual $q_r^t$, $u \in [1, M]$

12 $\quad$ **if** $\Delta > 0$ **then**

13 $\quad\quad$ $x_{ru}^t = x_{ru}^t - 1$

14 $\quad\quad$ $\Delta - = 1$

15 $\quad$ **end**

16 $\quad$ **else if** $\Delta < 0$ **then**

17 $\quad\quad$ $x_{ru}^t = x_{ru}^t + 1$

18 $\quad\quad$ $\Delta + = 1$

19 $\quad$ **end**

20 $\quad$ **else** $\Delta = 0$

21 $\quad\quad$ Break

22 $\quad$ **end**

23 **end**

---

## 4. Study Area and Data

### 4.1. Study Area

As the capital of Jiangsu Province, China, and the national gateway city for the central and western regions of the Yangtze River Delta, Nanjing is a world-famous historical and cultural city with an occupied area of 6587 km$^2$ and a population of 8,335,000 (Figure 4). Nanjing's medical and health system has been flourishing with relatively abundant comprehensive medical resources and overall health services that rank behind only Shanghai and Beijing in China. There are 241 public hospitals in Nanjing, of which 22 are top-tier 3A hospitals. As of 2019, Nanjing has 65 EMS stations (http://nanjing.emss.cn/newpages/show?id=969, accessed on 1 October 2020) that contain 169 emergency EMVs, characterized by high density in the central urban area and low density in the surrounding towns. Nanjing's emergency network depends on hospitals setting up several EMS stations with sufficient range to enable coordinated scheduling and distributed care. Nevertheless, this approach prevents EMVs from returning to the incident scene after the response is complete. The average emergency response time of Nanjing's emergency network is 16 min, which falls short of the national standard (less than 12 min)

and is much longer than developed countries' standards (less than 10 min) [48]. Given the existing situation of first aid, it is urgent to conduct optimization research on EMV relocation based on a time-varying environment in Nanjing. Therefore, we selected Nanjing as the research area in joint consideration of its population, transportation infrastructure, and medical services.

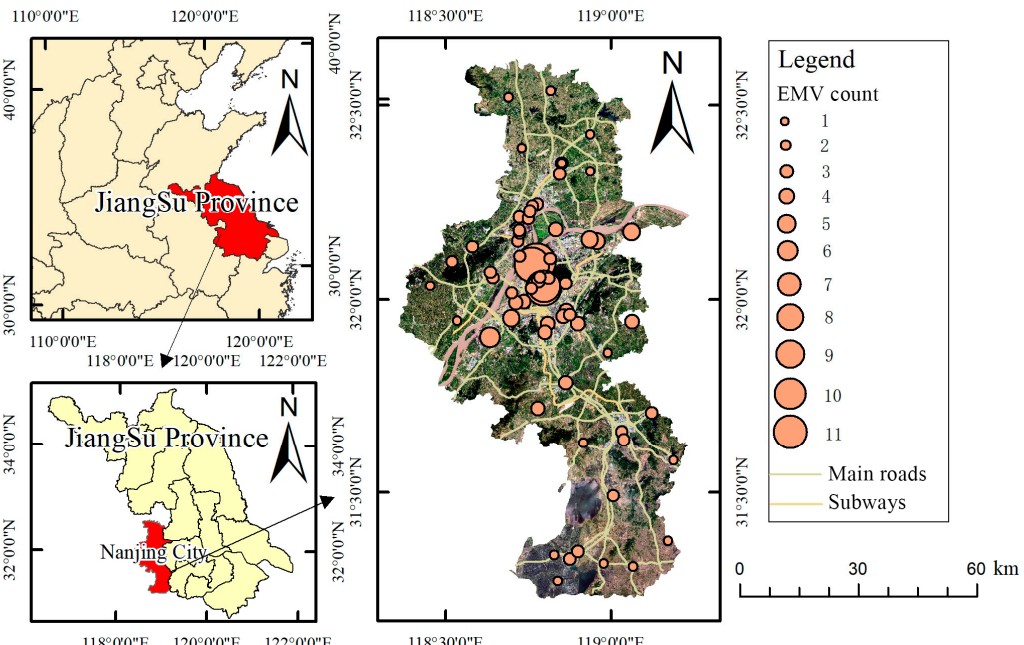

**Figure 4.** The study area.

*4.2. Data*

An efficient data acquisition system for summarizing data sufficiently is necessary when the EMVs are dynamically located [19]. Three types of data are collected from the internet: M-ORP data, high spatiotemporal resolution population data based on M-MPR, and EMS data to support EMV relocation optimization. A total of 870 community blocks are separated into three areas: central urban area, inner suburban area, and outer suburban area (The central urban area includes six districts: Gulou, Qinhuai, Qixia, Jianye, Yuhuatai, and Xuanwu; the inner suburban area refers to locations adjacent to the central urban area, including Jiangning and Pukou; the outer suburban area refers to areas far from the central urban area, including Gaochun, Lishui, and Liuhe). The potential commuting demand in downtown areas is high during the daytime and low at night [49,50], so we selected four typical time periods—8:00, 13:00, 18:00, and 22:00—to represent four time intervals: morning commuter peak, daytime commuter trough, evening commuter peak, and nighttime commuter trough, which can effectively manifest the impact of travel time and distance caused by commuter variation in the city. In a megacity such as Nanjing, the mobility of millions of residents produces a varying pattern of the population distribution over time.

4.2.1. Multiperiod Online Route Planner (M-ORP) Data

The web mapping route planning API is a feasible travel impedance calculation approach [51]. In this article, Amap Maps (Amap (www.amap.com, accessed on 1 October 2020) is one of the largest web map providers in mainland China, providing various LBS services) is selected as the data source. The route planning service of Amap Maps offers real-time navigation by producing tailor-made travel plans for users based on destination, departure, and path policy settings combined with real-time traffic [52]. As illustrated in Figure 5, we take each first aid station as the center to radiate to the surrounding areas, and the distance that can be reached in 30 min of driving is used as the threshold value. The

central urban area presents a prominent network structure. Because the first-aid stations in the central urban area are densely distributed and equipped with a relatively large number of EMVs, they show robust connectivity with the community. In the suburbs and outskirts, the first aid station is the center, and there are faults in the marginal zone. The most obvious are the marginal zones in the west and east of Jiangning, south of Lishui, Liuhe, and Gaochun. The examples in Figure 5b–e show considerable differences in traffic congestion from the emergency station of ZhongDa Hospital to the potential emergency site at different times, among which the morning and evening peaks are the most serious. Moreover, when a traffic accident occurs, the congestion is intensified. Therefore, M-ORP data must be used to characterize the dynamic changes in traffic commuting conditions.

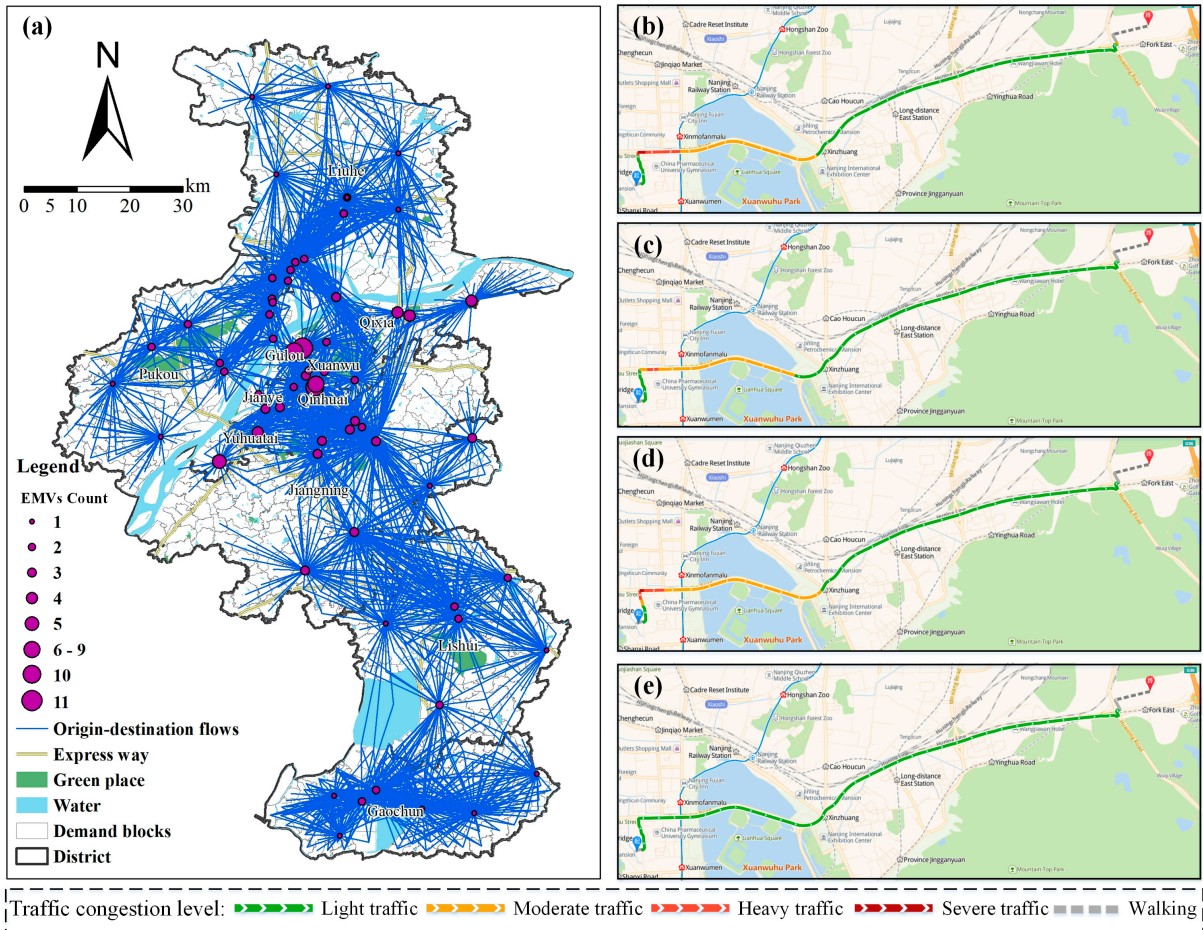

**Figure 5.** Diagram of M-ORP data. (**a**) each first aid station radiating to the surrounding areas within a 30-min driving threshold during the morning commuter peak. (**b**–**e**) examples of M-ORP data from ZhongDa Hospital to a potential emergency site at 8:00, 13:00, 18:00, and 22:00. (**b**–**e**) serve as a notable visual representation of varying degrees of traffic congestion, thereby exerting an influence on the overall duration of traffic.

### 4.2.2. Multiperiod Mobile Phone Records (M-MPR) Data

With the advancement of ICT in smart cities, large-scale transportation travel data such as mobile phone data, floating car data, and bus IC card data provide new channels for the description and understanding of urban areas and residents' behavior, and are increasingly applied in transportation planning and urban research [53,54]. This article utilizes the mobile phone signaling data of Nanjing in a particular week in the second half of 2018, which is relatively stable (as shown in Figure 6). The M-MPR data of Nanjing are obtained once every hour.

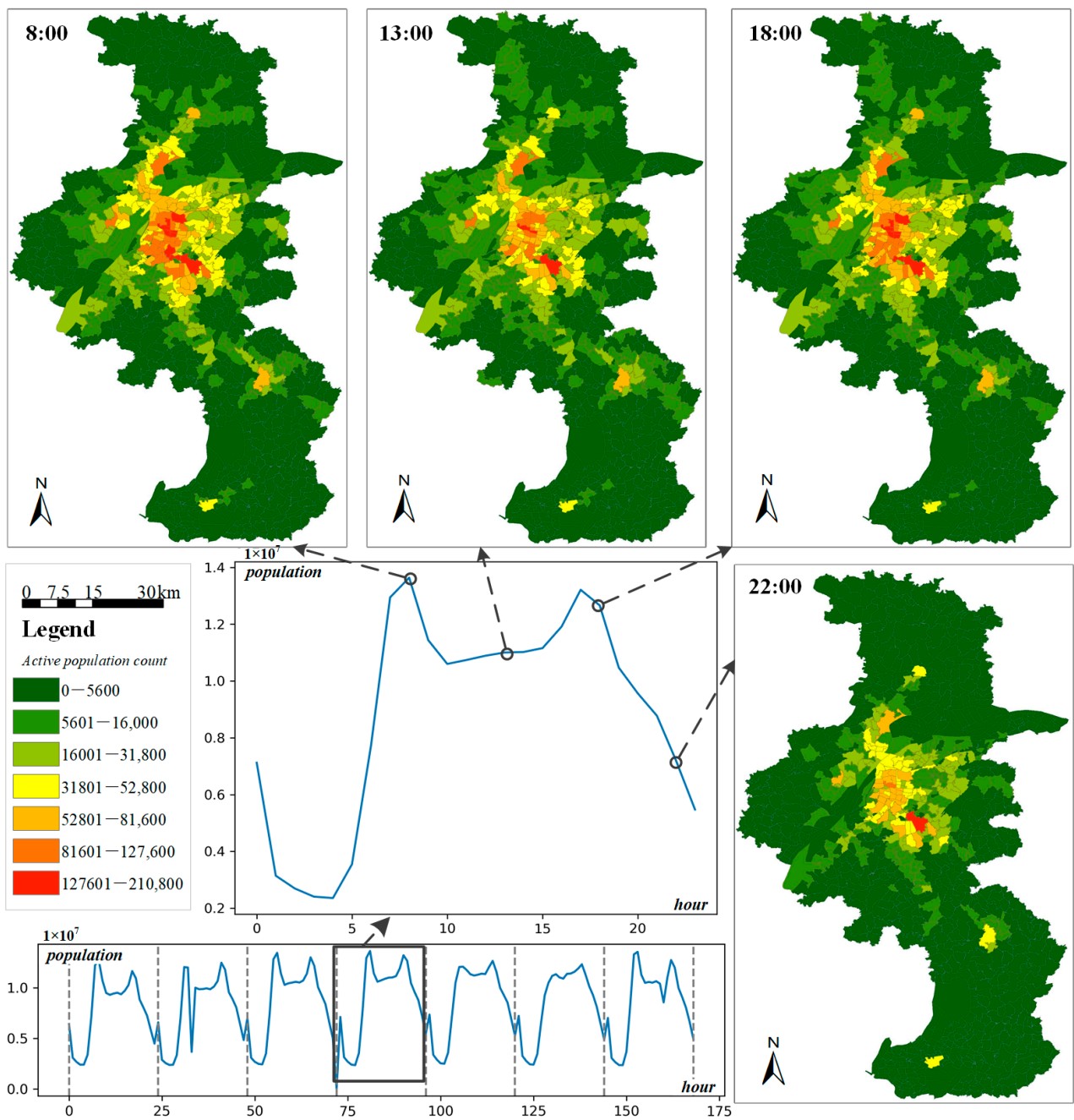

**Figure 6.** Distribution map of the population based on M-MPR data.

A person in need of EMSs must provide their current location, which is probably not the location of their home [1]. In the real urban space, personal travel activities—for example, employees on the way to a destination at a one-time slot and returning to their original location at another time slot—show a certain degree of mobility. As shown in Figure 6, the active population in Nanjing dynamically varies with time and place, a phenomenon that is most apparent in the main urban area. Approximately 41% of inpatients go to the hospital due to an emergency rather than a planned manner [55]. Therefore, the mobility of the population distribution must be considered when studying the spatiotemporal layout of EMSs. The total active population during the four time slots (8:00, 13:00, 18:00, and 22:00) is 12.94 million, 11.01 million, 12.65 million, and 7.20 million, respectively, corresponding to the actual active population in Nanjing.

## 5. Results

### 5.1. Relocation Experiment

We designed 15 group control experiments with different parameter settings, including iteration count, population size, crossover probability (*CP*), mutation probability (*MP*), crossover count (*CC*), mutation count (*MC*), *top*, and *bottom*. The experimental results were evaluated in terms of the average fitness and max fitness (Table 1). The objective fitness value in Table 1 is dimensionless quantity, represented as Equation (10) and calculated by the CIEA approach. The optimization search process of five group controls is illustrated in Figure 7. The result of the D1 group max fitness is the largest, as high as 23.38, when the *bottom* is not allowed to be 0, indicating that the crossover count and mutation count are set to 4. In contrast, when the *bottom* is permitted to be 0, the E3 group result achieves the maximum fitness. In the **E3** group, the divest-and-exit strategy was allowed to be adopted, and we observed significant improvement in the max fitness (42.41) and average fitness (19.99), which indicated that some emergency stations were redundant or should be relocated. Average fitness presented a convergence tendency, i.e., increasing initially and then oscillating. The maximum fitness presented a stable tendency after rising, indicating that the CIEA method showed better convergence and global search capability.

**Table 1.** Experimental parameter settings and results of the spatial evolution algorithm with integer constraints. A total of five control groups (A–E) were established, with each control group undergoing three experiments (1–3) employing the rigorous technique of variable control. The corresponding control variables can be referenced in the table. The travel time threshold (Equation (4)) is set to 0.5 h, and the $\beta$ coefficient (Equation (4)) is set to 1.

| Group | Iteration Count | Population Size | *CP* | *MP* | *CC* | *MC* | *top* | *bottom* | Average Fitness | Max Fitness |
|-------|-----------------|-----------------|------|------|------|------|-------|----------|-----------------|-------------|
| A1 | 250 | 200 | 0.8 | 0.2 | 8 | 8 | 15 | 1 | 10.54 | 19.47 |
| A2 | 500 | 200 | 0.8 | 0.2 | 8 | 8 | 15 | 1 | 9.9 | 18.86 |
| A3 | 1000 | 200 | 0.8 | 0.2 | 8 | 8 | 15 | 1 | 9.745 | 18.60 |
| B1 | 250 | 400 | 0.8 | 0.2 | 8 | 8 | 15 | 1 | 10.14 | 20.45 |
| B2 | 500 | 400 | 0.8 | 0.2 | 8 | 8 | 15 | 1 | 11.32 | 20.72 |
| B3 | 1000 | 400 | 0.8 | 0.2 | 8 | 8 | 15 | 1 | 11.20 | 19.25 |
| C1 | 500 | 200 | 0.9 | 0.1 | 8 | 8 | 15 | 1 | 11.87 | 21.61 |
| C2 | 500 | 200 | 0.7 | 0.3 | 8 | 8 | 15 | 1 | 10.00 | 17.22 |
| C3 | 500 | 200 | 0.6 | 0.4 | 8 | 8 | 15 | 1 | 7.31 | 17.24 |
| D1 | 500 | 200 | 0.8 | 0.2 | 4 | 4 | 15 | 1 | 14.45 | 23.38 |
| D2 | 500 | 200 | 0.8 | 0.2 | 12 | 12 | 15 | 1 | 8.81 | 16.82 |
| D3 | 500 | 200 | 0.8 | 0.2 | 16 | 16 | 15 | 1 | 6.99 | 15.91 |
| E1 | 500 | 200 | 0.8 | 0.2 | 8 | 8 | 20 | 1 | 11.72 | 21.15 |
| E2 | 500 | 200 | 0.8 | 0.2 | 8 | 8 | 10 | 1 | 11.14 | 19.32 |
| E3 | 500 | 200 | 0.8 | 0.2 | 8 | 8 | 10 | 0 | 19.99 | 42.41 |

### 5.2. Spatial Distribution Results of EMS after Optimization

We selected the **E3** group in Table 1, which allows the divest-and-exit strategy as a potential application scenario to lay out both the spatial distribution result and parallel coordinate system diagram at different time slots (Figure 8). As shown in Figure 8, the optimized dispatch volume of EMS stations in the central and inner suburban areas changed dramatically at different times. However, a slight increase was seen in the overall dispatch volume of EMS stations in the outer suburbs, and the existing partial stations were closed. Based on the characteristics of the changes, the relocation pattern of different EMS stations was divided into increasing patterns, fluctuating patterns, and descending patterns.

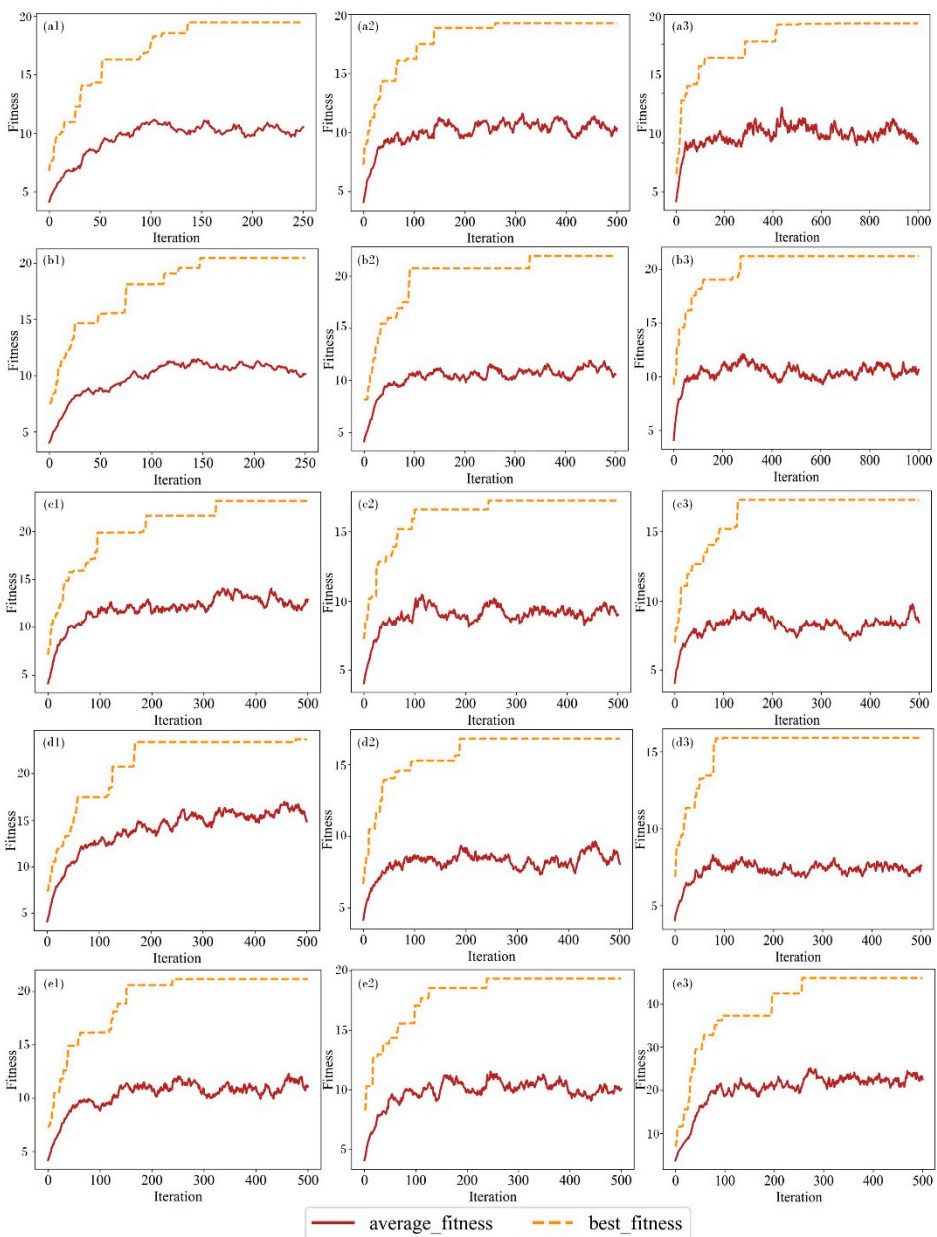

**Figure 7.** Experimental calculation convergence graphs of 15 groups for the CIEA method. The subfigures (**a1**–**a3**,**b1**–**b3**,**c1**–**c3**,**d1**–**d3**,**e1**–**e3**) are associated with their respective control groups (A1–E3), as delineated in Table 1.

(1) An increasing pattern means that the number of EMVs at each time slot increases significantly after optimization of the EMS station, indicating that the original resources of the EMS stations were insufficient. Only a few stations, such as Lishui and Qixia, fit this growth pattern perfectly. Moreover, the districts showing a significant increase at night include Pukou, Liuhe, and Jiangning, a trend that was directly related to the districts' large area, suburban location, and dense population.

(2) A fluctuating pattern refers to the number of EMVs fluctuating during each time slot with the dynamic changes in population and traffic conditions, indicating that the original resources of EMS stations roughly satisfied the demand. Furthermore, the dynamics after time-dependent optimization effects are reflected. Figure 8 shows that most of the sites, including Jianye, Pukou, Liuhe, and Xuanwu, fully conform to the fluctuation pattern.

(3) A descent pattern refers to the fact that the number of EMVs at each moment shows a significant decline after optimization of the EMS station, indicating that the original resource allocation of the EMS station easily satisfies the region's demand. Some stations in the central urban area, such as Qinhuai and Gulou, perfectly correspond to the descending pattern (Figure 8).

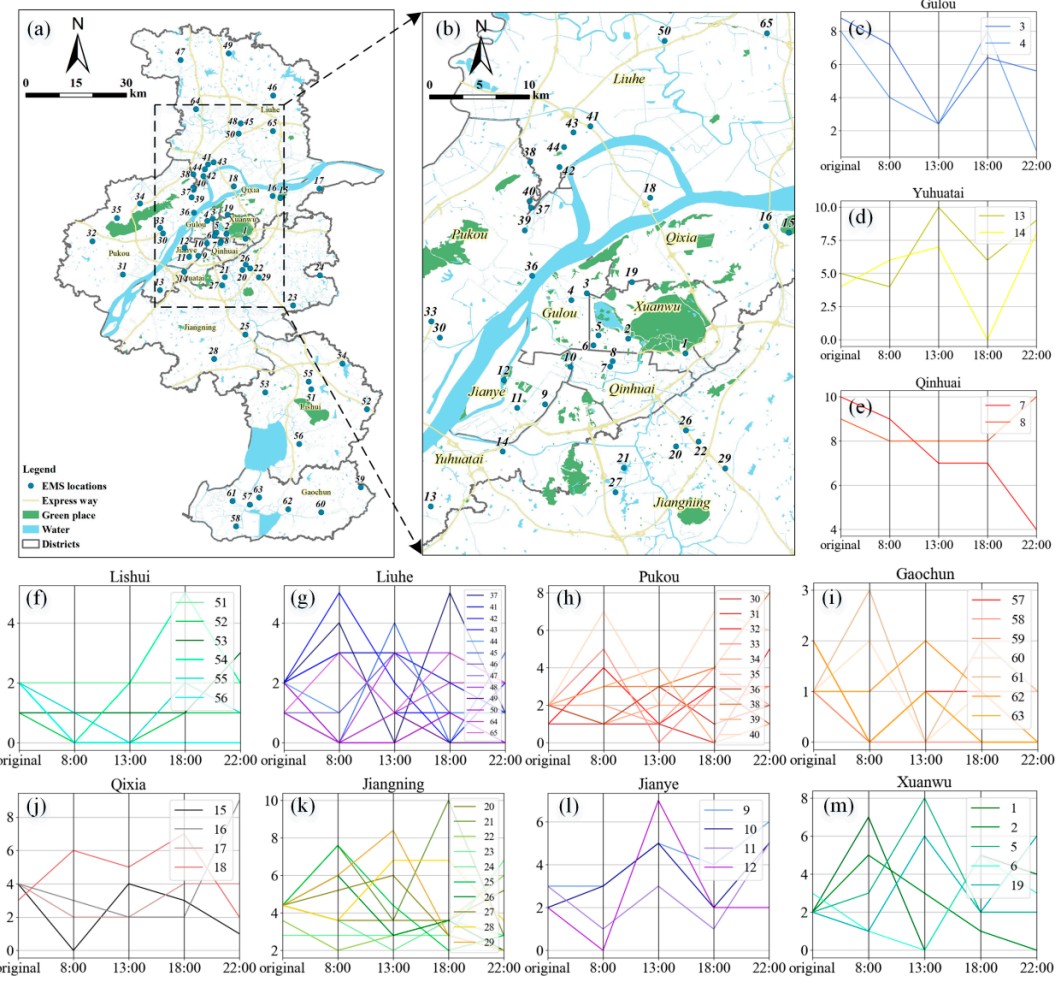

**Figure 8.** The parallel coordinate system diagrams after relocation optimization at four time slots. (**a**) ID of EMS stations in Nanjing, (**b**) zoom map of downtown part, (**c**–**m**) relocation strategy of each EMS station at four time slots after optimization in differ districts, refers to Gulou, Yuhuatai, Qinhuai, Lishui, Liuhe, Pukou, Gaochun, Qixia, Jiangning, Jianye, Xuanwu.

### 5.3. Equality Comparison before and after Optimization

This section involves the contrast indicators of accessibility disparities before and after optimization.

From the dimensions of temporal variation, the global equality values (The global equality is the sum of the equality values of all blocks during one time slot in the research area) before optimization were 5.924, 7.767, 8.138, and 15.595, implying that the time-dependent effect of EMS equality was significant. The global equality values after optimization were 8.943, 9.926, 13.078, and 57.187. Equality after optimization increased by 41.5% on average, and 33.7%, 21.7%, 37.7%, and 72.7% at different time slots. We calculated the equality day–night ratio (Equality day–night ratio = night equality value/daytime equality average) as 5.4 (before optimization) and 2.1 (after optimization).

From the dimension of spatial distribution (Figure 9), the central urban area (The central urban area includes six districts, i.e., Gulou, Qinhuai, Qixia, Jianye, Yuhuatai, and Xuanwu) is basically in the middle-value range before optimization, while the outer

suburban area (The outer suburban area refers to locations far from the central urban area, including Gaochun, Lishui, and Liuhe) is concentrated in the high- to very high-value range. The most typical representative areas include the urban areas of Gaochun and Lishui, which contradicts our stereotype. After optimization (Figure 10), the high-value regions were significantly reduced, and the high-value area in the central urban area expanded, typically during the evening commuter peak. The spatial distribution of the moderate- and low-value regions also expanded. These phenomena indicated that idle and redundant EMVs were dispatched to areas with lower equality, owing to the objective guides and CIEA optimization.

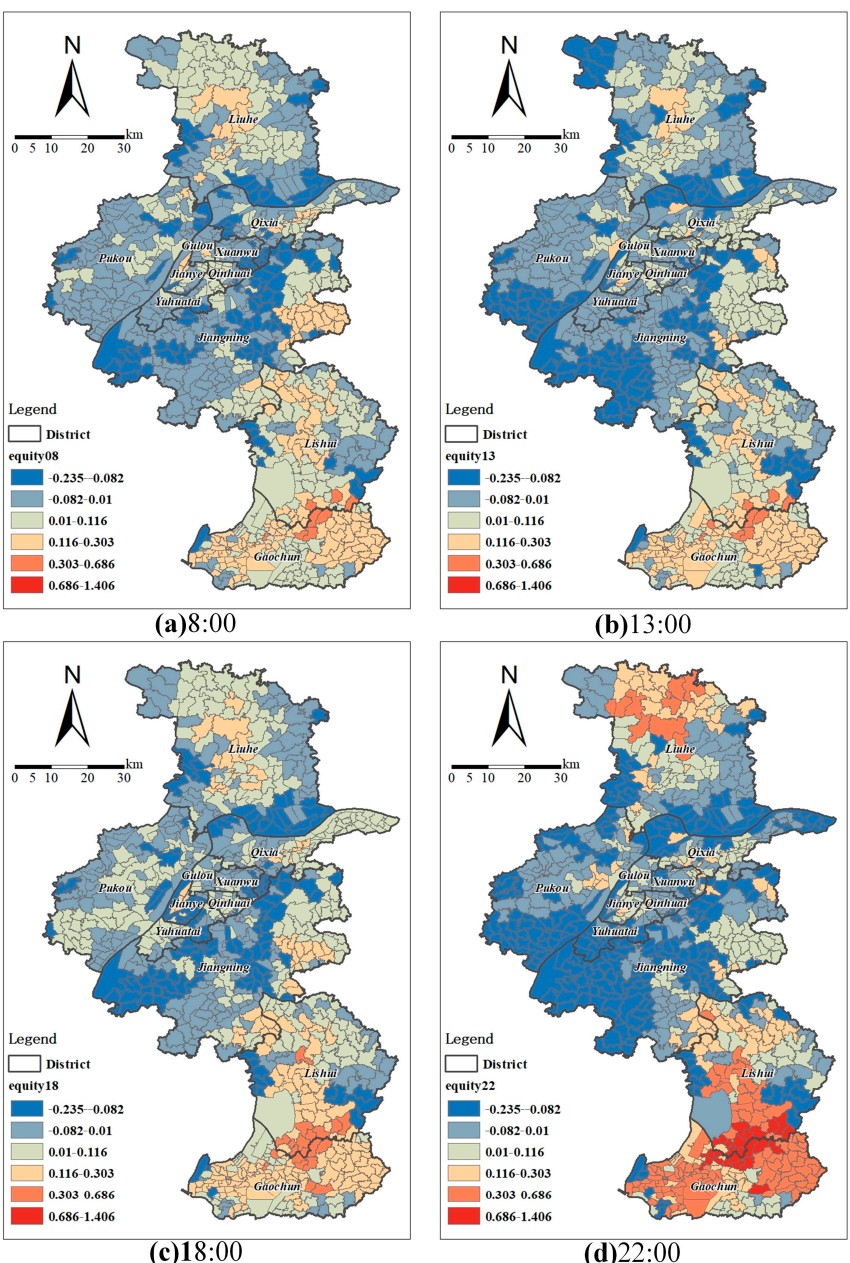

**Figure 9.** Spatial distribution maps of equality at different time slots before optimization. The spatial distribution maps of equality before optimization include (**a**) 8:00 (morning commuter peak), (**b**) 13:00 (daytime commuter trough), (**c**) 18:00 (evening commuter peak), and (**d**) 22:00 (nighttime commuter trough).

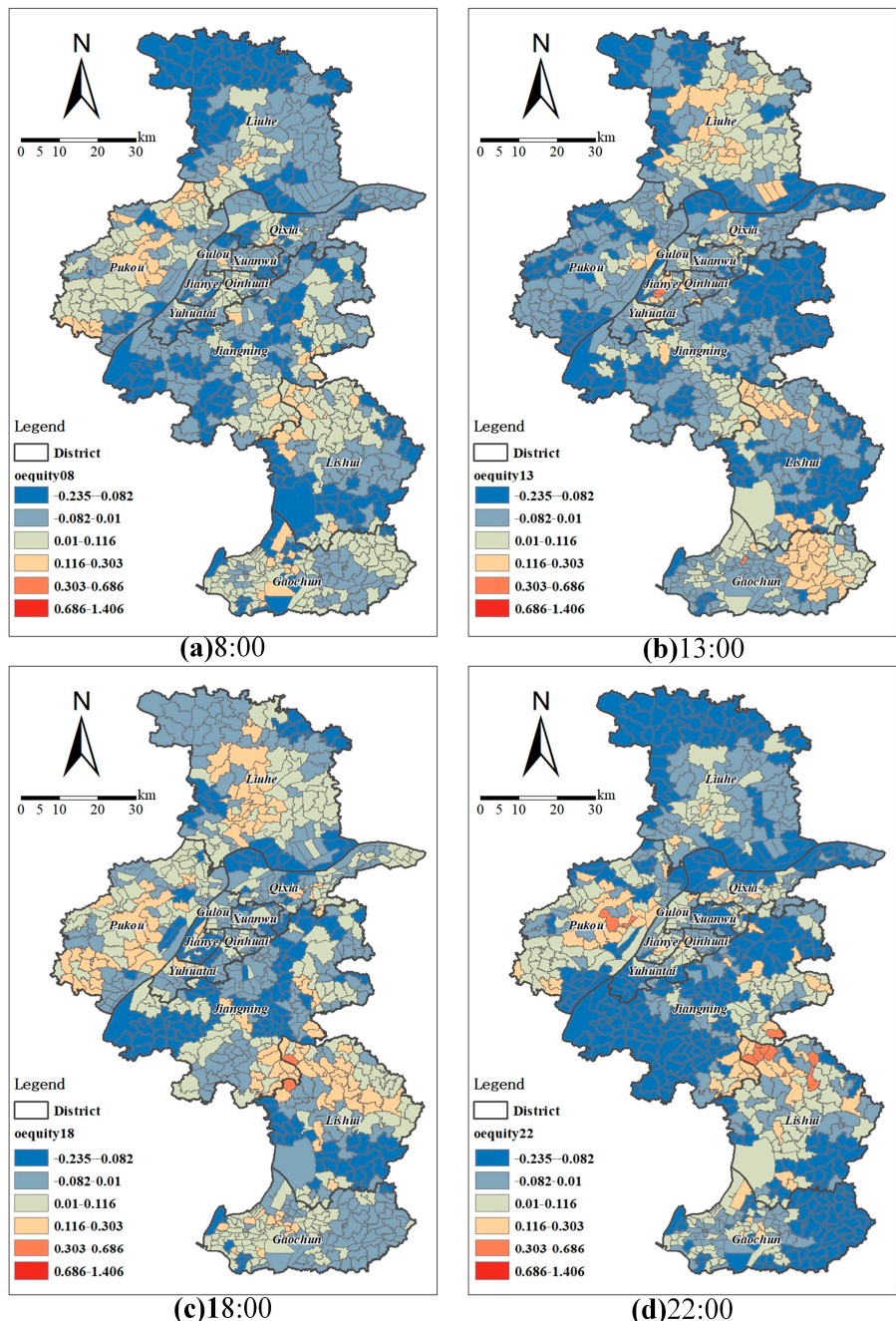

**Figure 10.** Spatial distribution maps of equality at different time slots after optimization. Additionally, the spatial distribution maps of equality after optimization include (**a**) 8:00, (**b**) 13:00, (**c**) 18:00, and (**d**) 22:00.

## 6. Discussion and Conclusions

### 6.1. Discussion

EMSs significantly impact modern health systems, and the timeliness of the response to emergency calls is of great importance to patient health and recovery [10]. Developing new reliable models (possibly hybrid models, supported by new ICT solutions) has become a primary test from which models can represent the inherent complexity [6].

The main contribution of this study is the presentation of the MRO-EMS approach based on multisource spatiotemporal big data. The proposed MRO-EMS framework innovatively integrates the equality objective function of time-dependent factors and the evolutionary operator of spatial characteristics. The research results exhibited applica-

ble examples of utilizing big spatiotemporal data for the quantified fine optimization of human–urban interactions, demonstrating promising advantages among various disciplines. The MRO-EMS approach focused on spatiotemporal equality, considering dynamic time dependence in the real world. The spatial equality of EMS shows high spatiotemporal sensitivity. As shown in the results, the equality of EMS in the research area increased by 41.5% after optimization, which could have a significant positive and incentivizing effect on alleviating the tension of prehospital medical facilities and promoting equality. By redeploying idle EMVs between stations and moving EMVs from potential low-demand stations to high-demand stations, the EMS dispatching center can couple the demand status of the urban population and traffic environment over various periods, ensure that the call demand near high-demand stations can be quickly responded to, and reduce the overall response time.

To further represent the numerical distribution's change, we generated a box plot (Figure 11) to compare the global equality value variation before and after optimization. Before optimization, the overall values are scattered, and the number of abnormally high values is significant. After optimization, the overall values are clustered around the mean, and the overall value decreases, reflecting the effects of the optimization approach. The unfavorable part after optimization must be noted. The low-value area has increased because the influence of the high-value, and very high-value areas on the objective function is far greater than that of the low-value area, which reflects the insensitivity of the objective function to the low-value area at a certain level.

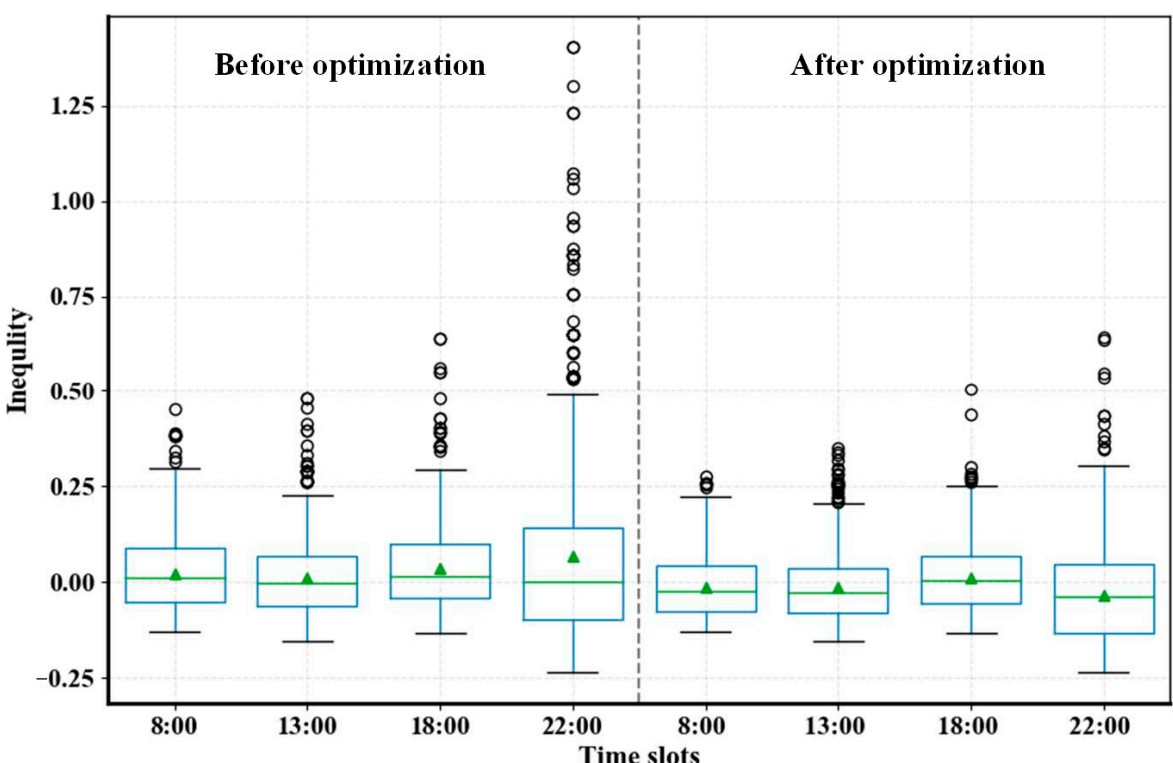

**Figure 11.** Box plot comparison before and after optimization.

The demand, the number of available vehicles, and travel times are all presumed to be changeable over time in the proposed models [10]. These two new data types use time-dependent data conditions to develop the dynamic time-space accessibility of EMSs. Realistic M-ORP data can dynamically characterize traffic congestion, presenting a key factor affecting EMS. Moreover, the distribution of the active population characterized by M-MPR data can reflect the distribution of the active population at different times in the entire region, even in the presence of data expansion and insufficient sampling errors. In a dynamic time-dependent environment, the CIEA is utilized to perform MRO-EMS,

thereby achieving maximum global equality within a day for a typical EMS, i.e., an EMV. Emergency demand is dynamic and time-varying; the dynamic EMV dispatch strategy can optimize emergency response capabilities under limited EMS resources. This article's research framework also applies to other time-sensitive service facilities, such as the location of stores, considering the time-dependent influence.

In terms of EMS organization types [56] (EMS organizations belong to two main groups, Anglo-American and Franco-German, defined by Dick), Nanjing, China, belongs to the Anglo-American group that is often not tied to the medical system and provides only nursing care with the intention of responding to calls as quickly as possible, and guaranteeing timely and safe transport of patients to appropriate medical facilities [10].

(1) EMSs in Nanjing show an aggregate developmental effect; that is, the Matthew effect [57]. However, the distribution of the central city is also uneven. Currently, the EMSs in the major city are concentrated, and the number of EMVs is considerable. Some social groups are concentrated in specific areas in the inner city and form localized contexts in which access to medical facilities and socio-institutional factors intersect and combine to create unequal accessibility. When a strong demand for rescue services close to these EMS stations exists, it is simple to create a shortage that can only be filled by stations located farther away, leading to a longer response time that has a detrimental effect on the overall average response time. As a result, we should monitor how these EMS stations operate from the perspective of the EMV dispatching center, construct more intense auxiliary stations to dispatch more vehicles on time, and increase the emergency network's resiliency.

(2) Echoing the triple jeopardy of social, environmental, and health inequalities [58], some rural-to-urban migrants may experience health-related disadvantages. This effect becomes significant after considering the spatial accessibility to medical facilities and spatial heterogeneity, suggesting the coupling of institutional and spatial factors over space, which is consistent with the findings of previous studies [59]. The current EMS is concentrated in areas where large hospitals are located, thereby making it possible to quickly solve the first aid station office space problem. After relocation optimization, EMVs should be dispatched to different districts of the central city.

(3) In addition, the reason some stations are not dispatched with EMVs and are closed in the potential application scenarios (Section 5.2) is that the selected EMS station locations may not be optimal; the same phenomenon has been observed in Amsterdam [27].

*6.2. Conclusions*

This research focuses on the operational decision-making process by solving the problem regarding dispatching and relocation of EMVs, and attempts to reconcile the discrepancy between the demand and provision of EMSs at various periods. We provided the MRO-EMS framework based on big spatiotemporal data under a restricted number of EMVs and limited scheduling scope. Additionally, we performed an empirical evaluation of real-world M-MPR and M-ORP data collected in Nanjing City, China, demonstrating the applicability of the MRO-EMS approach in large metropolitan areas. In addition, this framework can be extended to various service facilities, especially service facilities with strong time-dependent sensitivity.

Although this study reveals essential discoveries, there are also limitations. Therefore, several aspects of our further qualitative research work can be improved. First, the spatial unit is the street block, which is insufficient for high research accuracy of variable-scale research units, especially for the results of high time threshold sensitivity. Second, the model can be expanded to incorporate more realistic factors in EMV dispatch, such as fault handling, secondary dispatch, EMV heterogeneity, emergency station capacity limitations, and epidemic outbreaks such as SARS and COVID-19. Last, equality of accessibility is an essential objective, but service facility spatial allocation or relocation are complex multi-objective optimization problems. Therefore, we will design a multi-objective op-

timization framework to provide Pareto noninferior urban planning and management decision-making solutions.

**Author Contributions:** Xinxin Zhou designed the framework of this work and has drafted the work or substantively revised it. Yujie Chen analyzed and processed the spatiotemporal data, and was a major contributor in writing the manuscript. Bingjie Liu wrote the analysis algorithm. Yingying Li designed some figures and Zhaoyuan Yu was the instructor of this article. All authors have read and agreed to the published version of the manuscript.

**Funding:** This research was supported by the National Natural Science Foundation of China (Grant No. 42130103, 42201504, providing financial support), the Natural Science Research Start-up Foundation of Recruiting Talents of Nanjing University of Posts and Telecommunications (Grant No. NY221143, providing data support), and the Open Foundation of Key Lab of Virtual Geographic Environment of Ministry of Education (Grant No. 2021VGE02, providing data support).

**Data Availability Statement:** Restrictions apply to the availability of these data.

**Acknowledgments:** The authors would like to thank the anonymous referees and editor for their valuable comments, which significantly improved this paper.

**Conflicts of Interest:** The authors declare no conflict of interest.

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
