# Peer review of "Spatiotemporal Data-Driven Multiperiod Relocation Optimization of Emergency Medical Services: Maximum Equality Objective"

_ijgi, doi:10.3390/ijgi12070269_

Round 1
Reviewer 1 Report
I found this work interesting, potentially useful, and captivating to read.
There are, nevertheless, some issues that may need improvement, in my opinion:
1) In lines 407-408: “...so we selected four typical time slots, 8:00, 13:00, 18:00, and 22:00, to represent four time periods”
It is not clear if this represents measurements exactly at each hour mentioned or during some specific time interval, since the terms “slots” and “time periods” are used here. I suggest clarifying this better.
2) The map in Figure 5 seems quite “cluttered” in terms of cartographic design, so I suggest making it simpler or splitting it into more than one map. Also, the legend is not fully clear and should be more complete. For instance:
- what is the meaning of the values associated with the proportional circles – counts of EMSs, I guess?
- And in line 433: “(b)-(e) examples of MORP data from ZhongDa Hospital to potential emergency site at different times” – which times exactly do you mean?
- Also, the meaning of the colors associated with road segments in the (b)-(e) partial maps is not explained, and should be.
3) In lines 451-453: “The total active population during the four 451 time slots (8:00, 13:00, 18:00, and 22:00) is 1294, 1101, 1265, and 720, respectively, corresponding to the actual active population in Nanjing.” What do the values 1294, 1101, 1265, and 720 mean exactly? Counts of persons? And is there a multiplying factor involved, e.g., X1000?
4) In Figure 6, what are the values associated with color scale classes? The number of estimated inhabitants?
5) Under chapter 5.1, starting in line 458, I fail to understand the criteria/rationale for the definition/design of the 15 experimental groups. How were their (combination of) values chosen? Related to this, why are the groups (apparently) divided into types (A, B...), according to Table 2?
Author Response
Review Response
Dear Reviewer,
Thank you for your comments on our manuscript. We appreciate the time and effort you have taken to provide us with your feedback. We have carefully considered your comments and suggestions and have made the necessary revisions to our manuscript.
Here is the overall description of the revisions made to this article:
(1) The introduction section has been streamlined.
(2) The figures throughout the article have been further refined in terms of captions, explanations, and resolution enhancement.
(3) A comprehensive examination of the equations in the entire document has been conducted, with particular focus on variable definitions and explanations.
(4) The references have been renumbered according to the ACS style format.
(5) The description of the optimization algorithm has been formalized, highlighting the objectives, constraints, and innovations of the optimization algorithm.
(6) The content in the tables has been further explained or adjusted in terms of the positioning of the figure captions.
(7) A comprehensive reading and examination of the entire document were undertaken to ensure coherent logical progression and precise conveyance of meaning.
We have addressed each of your concerns in turn below:
Reviewer 1
Comment 1
In lines 407-408: “...so we selected four typical time slots, 8:00, 13:00, 18:00, and 22:00, to represent four time periods”. It is not clear if this represents measurements exactly at each hour mentioned or during some specific time interval, since the terms “slots” and “time periods” are used here. I suggest clarifying this better.
Response 1
This is a minor error. Thank you for bringing it to my attention. We have changed to time periods, rather than time slots.
Comment 2
The map in Figure 5 seems quite “cluttered” in terms of cartographic design, so I suggest making it simpler or splitting it into more than one map. Also, the legend is not fully clear and should be more complete. For instance:
- what is the meaning of the values associated with the proportional circles – counts of EMSs, I guess?
- And in line 433: “(b)-(e) examples of MORP data from ZhongDa Hospital to potential emergency site at different times” – which times exactly do you mean?
- Also, the meaning of the colors associated with road segments in the (b)-(e) partial maps is not explained, and should be.
Response 2
This suggestion is highly valuable and insightful. Thank you for sharing it with me! We also acknowledge the inherent cartographic ambiguity in representing the existing traffic flow, and thus, we attempted to decompose it into multiple maps for better visualization. However, due to the extensive experimental area and interdependencies among various administrative regions in terms of transportation connectivity, it proved challenging to separate the maps effectively. The primary purpose of Figure 5 (a) is to illustrate the relative strengths of transportation connections between different street blocks. Consequently, we opted to make appropriate improvements to the existing maps, and we sincerely hope for the understanding of domain experts. We have successfully incorporated legends in Figure 5 and 6 to augment the interpretive clarity of each timepoints.
Comment 3
In lines 451-453: “The total active population during the four 451 time slots (8:00, 13:00, 18:00, and 22:00) is 1294, 1101, 1265, and 720, respectively, corresponding to the actual active population in Nanjing.” What do the values 1294, 1101, 1265, and 720 mean exactly? Counts of persons? And is there a multiplying factor involved, e.g., X1000?
Response 3
Thank you for reminding me. We have revised and supplemented both the figures and the numerical values mentioned in the text to ensure precise and unambiguous representation.
Comment 4
In Figure 6, what are the values associated with color scale classes? The number of estimated inhabitants?
Response 4
Thank you for reminding me. As you said, The color scale classes of Figure 6 is the active number of estimated inhabitants. So we have made revisions to Figure 6 by incorporating additional explanatory legends and numerical axis labels.
Comment 5
Under chapter 5.1, starting in line 458, I fail to understand the criteria/rationale for the definition/design of the 15 experimental groups. How were their (combination of) values chosen? Related to this, why are the groups (apparently) divided into types (A, B...), according to Table 2?
Response 5
We have provided further explanation on the figures in Section 5.1, along with supplementary explanations regarding the interpretations of the horizontal and vertical axes. The grouping of the respective control groups remains distinct, as evident from the variations in conditions outlined in Table 2. It is noteworthy that the methodology employed in our study is based on the rigorous technique of controlling variables.
To sum up, we hope that these revisions address your concerns and that you find the revised manuscript acceptable for publication. If you have any further comments or suggestions, please do not hesitate to contact us.
Thank you again for your time and effort in reviewing our manuscript.
Sincerely, Xinxin Zhou

Reviewer 2 Report
This paper addresses the allocation of EMS service units to maximize the accessibility equality, considering the spatial-temporal variation of demand and travel costs. This paper proposes an objective function that measures the inequity of service accessibility and a capacitated integer evolution algorithm to allocate the EMS units. The approach is applied to an empirical study involving the ambulance allocation in Nanjing, China. The results shows that the equality is improved by 41.5% on average and the algorithm converges well after certain iterations. This work considers the spatial-temporal variability of main parameters, which is critical in assist the decision making in a real EMS planning problem.
Based on my review, I would suggest the authors to further work on follows:
- The current model/problem formulation needs significant amount of work.
- First, this paper deals with an optimization problem with certain constraints, it is hard for me to figure out the model formulation. It seems equation (10) - (17) formulates it. Please explicitly point out the main optimization problem that this paper address.
- Second, I would suggest the authors to have a thorough check to ensure all notations used in the paper are specified. For example, in equation (1), what is is Wij? What is T?
- Please have clear explanation on any notations. Examples that are confusing:
- It is hard for me to understand “E - the global provider scale” in Line 263 and “Q - the global demand scale” in Line 264.
- IEi is the inequality value of demand point i? Please make clear notation explanation. Also it is not a convention to discuss meaning of values in the notation specification. The extra discussion in Lines 259 - 262 “the higher the positive value is … “ can be put in the main text.
- Ai is already specified in line 225, no need to repeat it in line 257.
- When specifying constraints, some constraints are for each index, the authors should write this formally. For example, constraints (16) should be for each j.
- Another tip is to fix the index references. For example, the authors can specify i: index of demand point, j: index of service facility, t: index of time period, and avoid introducing other unnecessary indexes to increase the barrier for readers to understand. For example, index k is probably not needed in section 3.2.
- In equation (3), Ai is the accessibility of demand point i, so it should sum over all potential facility j, while j is summing over from 1 to N, which is the total number of demand points, should this for summing over j from 1 to M?
- In equation (7), the IEi is summing over i = 1 to M (the total number of supply points)?
- I do not understand equation (9), how the summation is equal to E divided by Q. Please explain.
- In equation (12), what is t’? Also, based on the formula, the delta should have two indexes t and t’, but there is not index. In equation (10), there is only the summation over t from 1 to T, but there should be another summation over t’.
- Equation (16), what is bottom? top?
- Overall, I would strongly suggest the author to check a few spatial optimization paper and learn how the optimization problem is commonly formulated in a standard and rigorous way.
- Algorithm:
- What is the main difference between the proposed CIEA and the normal EA algorithm? What is tailored for this specific problem?
- Can the author explain how capacity constraint is held in the cross over and mutation stages.
- Figure 3 (a) there are two 18o’ clock? I suppose the authors would like to shows four different time periods. Figure 3 (b) the left chromosome is the exactly same as the right one?
- Lines 354 - 355, I do not understand this sentence. How Zi is used.
- There seems a bug in the algorithm in Table 1: delta should be specified and initialized at the beginning of the algorithm, otherwise, it could possibly be undefined when going to line 10, if line 7 is not satisfied.
- What is Xru? How to pick which X for incrementing or decrementing between lines 10 and 23.
Miscellaneous:
- The figures in the paper has low resolution quality, please use better quality figures.
- When referencing equations, please be consistent, for example using Eq. (1) instead of Eq. 1.
- Please align equations and main texts, it is observed when there is an equation, the texts and equations are off.
- Many figure / table captions contains detailed description (e.g., Figure 2, Figure 3, Table 2, etc.). Please move them to the main text.
- Line 346 there seems missing a parenthesis.
- There is Chinese in Table 1…
- Figure 6, please add axis title to the chart.
- Line 517 “the spatial distribution of the high-value regions was significantly reduced” I do not understand this sentence.
Author Response
Review Response
Dear Reviewer,
Thank you for your comments on our manuscript. We appreciate the time and effort you have taken to provide us with your feedback. We have carefully considered your comments and suggestions and have made the necessary revisions to our manuscript.
In order to provide a coherent and comprehensive response to the points raised in your review, we have meticulously organized and addressed each of your comments individually.
Here is the overall description of the revisions made to this article:
(1) The introduction section has been streamlined.
(2) The figures throughout the article have been further refined in terms of captions, explanations, and resolution enhancement.
(3) A comprehensive examination of the equations in the entire document has been conducted, with particular focus on variable definitions and explanations.
(4) The references have been renumbered according to the ACS style format.
(5) The description of the optimization algorithm has been formalized, highlighting the objectives, constraints, and innovations of the optimization algorithm.
(6) The content in the tables has been further explained or adjusted in terms of the positioning of the figure captions.
(7) A comprehensive reading and examination of the entire document were undertaken to ensure coherent logical progression and precise conveyance of meaning.
Reviewer 2
We have addressed each of your concerns in turn below:
Comment 1
First, this paper deals with an optimization problem with certain constraints, it is hard for me to figure out the model formulation. It seems equation (10) - (17) formulates it. Please explicitly point out the main optimization problem that this paper address.
Response 1
As you have correctly comprehended, the primary optimization problem is described by Equations (10) to (17). In order to provide clearer elucidation, we have made modifications to the descriptions in the manuscript, particularly in Sections 3.3 and 3.4, focusing on the objective function, the meaning of key parameters, and boundary constraints.
Comment 2
Second, I would suggest the authors to have a thorough check to ensure all notations used in the paper are specified. For example, in equation (1), what is is Wij? What is T? Please have clear explanation on any notations.
Response 2
We appreciate your astute observation regarding this evident oversight. In response, we have supplemented the explanation of the relevant variables in Equation (1) and provided proper citation for its reference paper.
Comment 3
Examples that are confusing: It is hard for me to understand “E - the global provider scale” in Line 263 and “Q - the global demand scale” in Line 264.
Response 3
Thank you for your suggestion. We have revised the description accordingly. The precise expression is as follows:
- the total service (provider) supply capacity;
- the total demand amount.
Comment 1
IEi is the inequality value of demand point i? Please make clear notation explanation. Also it is not a convention to discuss meaning of values in the notation specification.
Response 4
As you have comprehended, we have provided explanations within the text and made concise revisions to the main body of the manuscript.
Comment 5
The extra discussion in Lines 259 - 262 “the higher the positive value is … “ can be put in the main text.
Response 5
We have condensed the content of this section in order to enhance conciseness and clarity.
Comment 6
Ai is already specified in line 225, no need to repeat it in line 257.
Response 6
Thank you for your reminder. We have promptly removed the redundant explanations.
Comment 7
When specifying constraints, some constraints are for each index, the authors should write this formally. For example, constraints (16) should be for each j.
Response 7
This suggestion is highly meaningful, and we have made modifications to the equation accordingly.
Comment 8
Another tip is to fix the index references. For example, the authors can specify i: index of demand point, j: index of service facility, t: index of time period, and avoid introducing other unnecessary indexes to increase the barrier for readers to understand. For example, index k is probably not needed in section 3.2.
Response 8
We extend our appreciation for your valuable suggestion. In response, we have included supplementary explanations concerning the frequently indexed variables. Further elucidation is warranted in this context, as the variable 'k' denotes values within a defined distance interval. is the travel cost from to any service site within the catchment, and is the catchment size. The detail computation of the Gaussian weights will be discussed in paper [1].
Comment 9
In equation (3), Ai is the accessibility of demand point i, so it should sum over all potential facility j, while j is summing over from 1 to N, which is the total number of demand points, should this for summing over j from 1 to M?
Response 9
Here, we are providing an explanation of a fundamental concept in 3SFCA. We have also confirmed that the summation range for 'j' should be from 1 to N, as stated in the primary reference article [1].
Comment 10
In equation (7), the IEi is summing over i = 1 to M (the total number of supply points)?
Response 10
The objective here is to minimize the disparities (inequities) in accessibility among all supply points, and thus we sum them up accordingly.
Comment 11
I do not understand Equation (9), how the summation is equal to E divided by Q. Please explain.
Response 11
This has been previously demonstrated in the studies of earlier scholars, and our paper directly adopts their findings. Wang and Tang [2] proposed an optimization model to minimize the difference in accessibility from demand points to provider locations, thereby realizing the maximization of equality, in other words, minimizing inequality.
Comment 12
In equation (12), what is t’? Also, based on the formula, the delta should have two indexes t and t’, but there is not index. In equation (10), there is only the summation over t from 1 to T, but there should be another summation over t’.
Response 12
Explanations have been added for variables such as t', and the role and intent of the cardinal sign function sgn(∆) have been further elucidated and clarified in the main body of the text.
- the lower (bottom) limit at provider ;
- the upper (top) limit at provider ;
- the total demand amount at time slot ;
- the total service (provider) resource capacity at time slot ;
- the original initial moment, representing the status quo before optimization;
- the accessibility value of demand at initial moment ;
- the global average value of accessibility at time slot ;
- the global average value of accessibility at initial moment .
Comment 13
Equation (16), what is bottom? top?
Response 13
Thank you for your suggestion. We have already provided supplementary explanations for the variables.
Comment 14
Overall, I would strongly suggest the author to check a few spatial optimization paper and learn how the optimization problem is commonly formulated in a standard and rigorous way.
Response 14
We extend our sincere appreciation for your valuable suggestion. In response, we have undertaken a thorough examination of the optimization objective, constraint conditions, and the optimization algorithm, ensuring their utmost clarity and accuracy in conveying the intended meaning.
Comment 15
Algorithm: What is the main difference between the proposed CIEA and the normal EA algorithm? What is tailored for this specific problem?
Response 15
Thank you for your suggestion. We have already provided supplementary explanations for the CIEA method.
Comment 16
Can the author explain how capacity constraint is held in the cross over and mutation stages.
Response 16
This heavily relies on the Capacitated Adjustment Operator. As observed from Figure 2, it is evident that after performing the crossover and mutation operations, the Capacitated Adjustment Operator is required to satisfy the constraint.
Comment 17
Figure 3 (a) there are two 18o’ clock? I suppose the authors would like to shows four different time periods. Figure 3 (b) the left chromosome is the exactly same as the right one?
Response 17
Thank you for your reminder. We have conducted a comprehensive revision of Figure 3.
Comment 18
Lines 354 - 355, I do not understand this sentence. How Zi is used.
Response 18
This expression is redundant. In order to reduce ambiguity, we have removed .
Comment 19
There seems a bug in the algorithm in Table 1: delta should be specified and initialized at the beginning of the algorithm, otherwise, it could possibly be undefined when going to line 10, if line 7 is not satisfied.
Response 19
represents the discrepancy between the total quantity after genetic operations and the specified . This explanation has also been included in the declaration of Table 1.
Comment 20
What is Xru? How to pick which X for incrementing or decrementing between lines 10 and 23.
gene loci sequence, denoted as {X1,X2... XnT}
Response 20
Here is an additional explanation: They represent the genetic sequences of the chromosomes, indicating the quantity of Emission Monitoring Vehicles (EMVs) at each site.
Comment 21
Miscellaneous:
- The figures in the paper has low resolution quality, please use better quality figures.
- When referencing equations, please be consistent, for example using Eq. (1) instead of Eq. 1.
- Please align equations and main texts, it is observed when there is an equation, the texts and equations are off.
- Many figure / table captions contains detailed description (e.g., Figure 2, Figure 3, Table 2, etc.). Please move them to the main text.
- Line 346 there seems missing a parenthesis.
- There is Chinese in Table 1…
- Figure 6, please add axis title to the chart.
- Line 517 “the spatial distribution of the high-value regions was significantly reduced” I do not understand this sentence.
Response 21
Thank you for your meticulous review. We have made revisions addressing these detailed issues and minor errors. Modification details are as follows.
(1) We conducted a resolution check on the primary figures in the manuscript and identified insufficient resolution in Figure 9 and Figure 10. We performed map generation and replacements accordingly.
(2) We replaced all formula citations throughout the document with the format "Eq. (1)."
(3) We finalized the typesetting and alignment.
(4) We consulted recent publications in IJGI to ascertain the preferred conventions for graphic and tabular descriptions. We observed variations in both title and main text for providing detailed descriptions. After a comprehensive evaluation, we restructured the analysis and procedural descriptions of phenomena in Table 2 and Figure 2 into the main text.
(5) We added necessary parentheses.
(6) We conducted thorough proofreading and inspection.
(7) We made modifications to the sentence structure in line 517.
We hope that these revisions address your concerns and that you find the revised manuscript acceptable for publication. If you have any further comments or suggestions, please do not hesitate to contact us.
Thank you again for your time and effort in reviewing our manuscript.
Sincerely, Xinxin Zhou

Reviewer 3 Report
Overall, this article journal is good and can be accepted for the journal after authors improve as comment state below:
1. Article Title, Abstract and Keyword - accepted
2. Introduction: Discussion on this subtopic is too long and certain item discusses in this subtopic can be move to subtopic Related Work or Methodology.
-. References, should not put the author names, is good if author can state as [1] etc. Example (M. Li, Wang, Kwan, Chen, & Wang, 2022). should be [1]
- The sequence of your references should be [1]/[2]/[3] etc and not just put wherever you want. It makes user don't want to read your article.
- In your article I tried to find M. Li, Wang, Kwan, Chen, & Wang, 2022 and I can't find this reference in your article except Li, M. Need author to check back.
3. Method: Authors are required to confirm the formatting especially line 218 and below, please look back and need to correct it.
4. References: Good to follow normal formatting that started with [1] and follows.
Author Response
Review Response
Dear Reviewer,
Thank you for your comments on our manuscript. We appreciate the time and effort you have taken to provide us with your feedback. We have carefully considered your comments and suggestions and have made the necessary revisions to our manuscript.
Here is the overall description of the revisions made to this article:
(1) The introduction section has been streamlined.
(2) The figures throughout the article have been further refined in terms of captions, explanations, and resolution enhancement.
(3) A comprehensive examination of the equations in the entire document has been conducted, with particular focus on variable definitions and explanations.
(4) The references have been renumbered according to the ACS style format.
(5) The description of the optimization algorithm has been formalized, highlighting the objectives, constraints, and innovations of the optimization algorithm.
(6) The content in the tables has been further explained or adjusted in terms of the positioning of the figure captions.
(7) A comprehensive reading and examination of the entire document were undertaken to ensure coherent logical progression and precise conveyance of meaning.
Reviewer 3
We have addressed each of your concerns in turn below:
Comment 1
Article Title, Abstract and Keyword – accepted
Response 1
Thank you for acknowledging my work.
Comment 2
Introduction: Discussion on this subtopic is too long and certain item discusses in this subtopic can be move to subtopic Related Work or Methodology.
-. References, should not put the author names, is good if author can state as [1] etc. Example (M. Li, Wang, Kwan, Chen, & Wang, 2022). should be [1]
- The sequence of your references should be [1]/[2]/[3] etc and not just put wherever you want. It makes user don't want to read your article.
- In your article I tried to find M. Li, Wang, Kwan, Chen, & Wang, 2022 and I can't find this reference in your article except Li, M. Need author to check back.
Response 2
Firstly, we have addressed this issue by making revisions and streamlining the introduction section.
Secondly, we have addressed the issue with the reference format and numbering sequence by conducting a comprehensive review and making revisions. The current reference style is MDPI ACS Style.
Thirdly, we have conducted further verification on the reference (M. Li, Wang, Kwan, Chen, & Wang, 2022) you mentioned. The reference titled (25. Li, M., Wang, F., Kwan, M.-P., Chen, J., & Wang, J. (2022). Equalizing the spatial accessibility of emergency medical services in Shanghai: A trade-off perspective. Computers, environment and urban systems, 92, 101745. doi:https://doi.org/10.1016/j.compenvurbsys.2021.101745.) is one of the influential and cutting-edge articles currently available on EMS optimization research. The authors are renowned experts in this field.
Comment 3
Method: Authors are required to confirm the formatting especially line 218 and below, please look back and need to correct it.
Response 3
We further examined the template format of the entire document and made corresponding alignment modifications.
Comment 4
References: Good to follow normal formatting that started with [1] and follows.
Response 4
we have addressed the issue with the reference format and numbering sequence by conducting a comprehensive review and making revisions. The current reference style is MDPI ACS Style.
We hope that these revisions address your concerns and that you find the revised manuscript acceptable for publication. If you have any further comments or suggestions, please do not hesitate to contact us.
Thank you again for your time and effort in reviewing our manuscript.
Sincerely, Xinxin Zhou

Round 2
Reviewer 2 Report
Thanks for addressing my comments.
Some minor items:
1. Formula (12) should be for each t? Maybe Delta should be specified as Delta_t (with a subscript t)
2. Formula (15) should be for each t
Author Response
Review Response
Dear Reviewer,
Thank you for your comments on our manuscript. We appreciate the time and effort you have taken to provide us with your feedback. We have carefully considered your comments and suggestions and have made the necessary revisions to our manuscript.
Reviewer
Comment 1
Formula (12) should be for each t? Maybe Delta should be specified as Delta_t (with a subscript t)
Response 1
Thank you for your suggestion. We have already added subscript t and provided supplementary explanations for the variables.
Comment 2
Formula (15) should be for each t.
Response 2
We appreciate your observation regarding this evident oversight. This suggestion is highly meaningful, and we have made modifications to the equation accordingly.
We hope that these revisions address your concerns and that you find the revised manuscript acceptable for publication. If you have any further comments or suggestions, please do not hesitate to contact us.
Thank you again for your time and effort in reviewing our manuscript.
Sincerely, Xinxin Zhou